# Ocean deoxygenation after the Sturtian Snowball

Kun Zhang [1] ✉, Susan H. Little [1], Alexander J. Dickson [2] & Graham A. Shields [1]

The abrupt ending of the Sturtian 'Snowball' glaciation was characterised by enhanced chemical weathering and carbon cycle perturbations, but there is less certainty over how oxygen levels responded to those changes. Here we reconcile conflicting views using a carbonate-based multiproxy dataset from the Taishir Formation in Mongolia. The geochemical data reveal an episode of ocean deoxygenation, followed by a shift toward less reducing, but still largely anoxic conditions in a post-glacial ocean characterised by nutrient and sulfate limitation. Ocean redox dynamics and biogeochemical cycling following the Sturtian deglaciation were likely dictated by unique tectonic and climatic regimes that facilitated the buildup of a recalcitrant dissolved organic carbon pool in the deep ocean. Post-glacial eutrophication may help to explain the delayed diversification of algal clades, but the persistence of ocean anoxia, excepting transient oxidation pulses, likely hindered the emergence of obligate aerobes, such as animals, until the Ediacaran Period.

The Cryogenian Period (c. 717 to 635 million years ago, or Ma) encompasses the Sturtian and Marinoan 'Snowball Earth' glaciations[1], together with an intervening non-glacial interlude during which a seemingly 'normal' hydrological cycle returned. The period represents a fundamental turning point in Earth-Life evolution, linking the 'Tonian Transformation'[2] with the 'Ediacaran-Cambrian Radiation'[3]. The non-glacial interlude (c. 661–650 Ma) is of particular interest due to it being a key stage in the rise toward ecosystem dominance by multicellular eukaryotes in the form of both algae and animals (putative sponges), as evidenced by organic biomarkers[4,5]. This ecosystem reshaping was accompanied by significant perturbations to the global carbon cycle as indicated by a high carbonate carbon isotope ($\delta^{13}C_{carb}$) baseline punctuated by transient negative $\delta^{13}C_{carb}$ excursions. Examination of the Cryogenian carbon cycle has raised the hypothesis that decreased organic carbon remineralisation associated with sulfate-poor deep ocean conditions led to the growth of a large dissolved organic carbon (DOC) pool[6], but the evidence remains controversial[7]. High $\delta^{13}C_{carb}$ values have also been related to ocean-atmosphere oxygenation, possibly caused by increased organic carbon burial and corresponding oxygen release due to a surplus of glacially induced nutrient supply, thus providing a potential link to early animal evolution[8]. Recent studies highlight the role of uninhabitable environments for limiting animal evolution, specifically pinpointing the immediate aftermath of the Sturtian deglaciation as one such key interval[9,10].

Proxy evidence and modelling studies point to enhanced volcanic activity[10], a brief burst of chemical weathering[11,12] and global cooling following a super-greenhouse climate[9,13] in the aftermath of the Sturtian deglaciation. A short-lived euxinic interval has also been inferred from iron speciation data[9], but the underpinning mechanisms are not fully understood. Moreover, while Mo isotopes from siliciclastic deposits support extensive marine euxinia[14,15], this would appear to contradict U isotope records from carbonate rocks that are interpreted to indicate transient ocean oxygenation[16]. These conflicting views hamper our understanding of mechanistic links between climate, ocean dynamics and biological evolution during such a critical interval.

In order to resolve these uncertainties, we revisit the Taishir Formation in Mongolia, which represents one of the few continuous successions deposited on an open carbonate ramp during the non-glacial Cryogenian interval[12] (Fig. 1). Details of the geological setting are provided in the Supplementary Information. We focus on well-preserved carbonate samples from the lowermost interval of the

[1]Department of Earth Sciences, University College London, London, UK. [2]Centre of Climate, Ocean and Atmosphere, Department of Earth Sciences, Royal Holloway University of London, Egham, Surrey, UK. ✉e-mail: kun-zhang@ucl.ac.uk

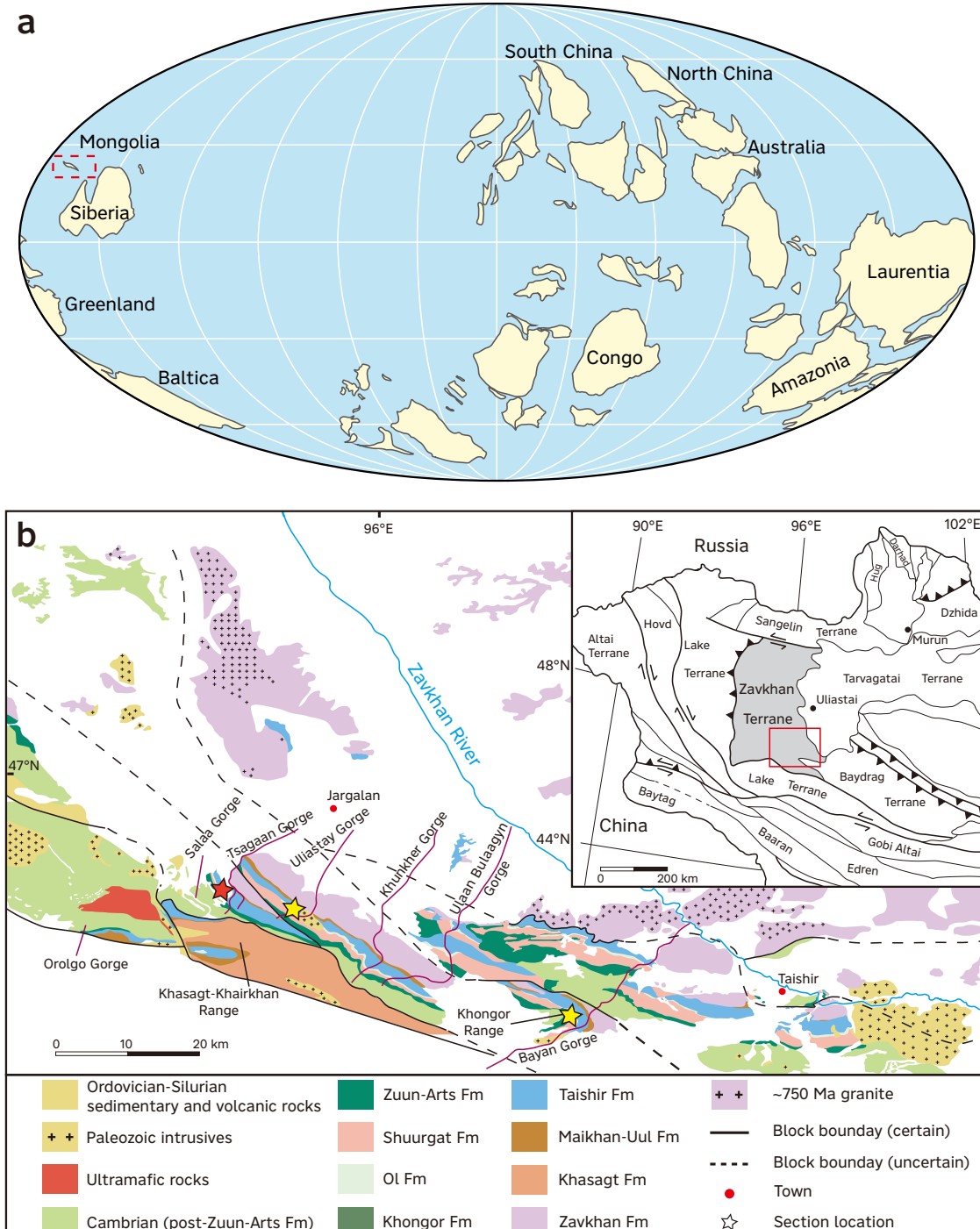

**Fig. 1 | Paleogeographic and geological map of the study area. a** Simplified paleogeographic reconstruction for the Cryogenian nonglacial interlude (modified after ref. [9]). Dashed rectangle indicates the possible location of Mongolia from ref. [9]. **b** Geological map of the Zavkhan Terrane in western Mongolia with the inset showing tectonic terrane map of western Mongolia (modified after ref. [16] with permission from Elsevier). Fm – Formation. The studied section is denoted by the red star while previously studied sections are shown as yellow stars.

Taishir Formation, which was deposited over several million years following the Sturtian Snowball deglaciation[11,17] (Supplementary Information). To attain insight into carbon and sulfur cycling, nutrients, weathering and marine redox conditions, we investigate a wide range of carbonate-based geochemical proxies that include carbon, sulfur, zinc, strontium and uranium isotopes, and rare earth elements (REE), phosphorus and iodine concentrations. These proxies are extracted from samples with high carbonate contents following established leaching protocols, and no significant contamination from non-carbonate phases is identified (Methods and Supplementary Information).

One major assumption in such chemostratigraphic studies is that geochemical proxies have regional or even global significance due to the chemical and isotopic homogeneity of the world's oceans. However, transient heterogeneity may have occurred in the immediate aftermath of Cryogenian glaciations due to the abrupt release of a meltwater plume, as evidenced perhaps by the relatively radiogenic $^{87}Sr/^{86}Sr$ ratios of carbonates overlying Sturtian diamictites in NW

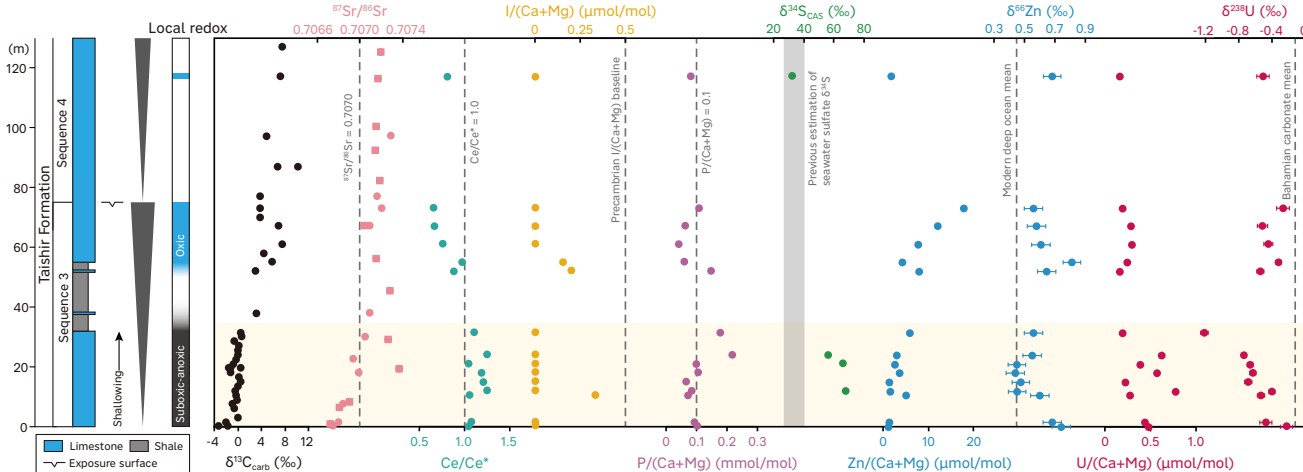

**Fig. 2 | Geochemical data profiles for the studied succession from the lower part of Taishir Formation at Tsagaan Gorge.** Sequence stratigraphy is from ref. 98. Lithostratigraphy, $\delta^{13}C_{carb}$ and $^{87}Sr/^{86}Sr$ data (circles) are from ref. 11. $^{87}Sr/^{86}Sr$ data (squares) of ref. 12 are correlated based on the age model of ref. 9. The Precambrian I/(Ca+Mg) baseline is from ref. 99. Estimation of seawater sulfate $\delta^{34}S$ is from ref. 47. The mean $\delta^{66}Zn$ value of the modern deep ocean is from ref. 53 and the mean $\delta^{238}U$ value of Bahamian carbonates is from ref. 100. The shaded band highlights the lower part of the succession. Note that I/(Ca+Mg) values below the detection limit are regarded as zero, and errors for some measurements of $\delta^{238}U$ are smaller than the data symbols.

Canada[18]. Sr isotope values in the studied interval are not anomalous but are instead consistent with the least radiogenic values from Twitya[18] and Rasthof[19] carbonates. Together with the lack of typical cap carbonate sedimentary features, the studied interval seems unlikely to have been affected by a meltwater plume during deposition (Supplementary Information). Additionally, petrographic observations and critical scrutiny of diagenetic effects (Supplementary Information) show that the first-order stratigraphic trends (Fig. 2) are indicative of changes in seawater composition, and have not been significantly overprinted by diagenetic processes. Therefore, these records enable us to depict a more holistic picture of post-Sturtian environments and interrogate potential links between environmental change and biological evolution during the rise to organic complexity.

## Results and discussion
### Local redox conditions
Cerium anomalies archived in carbonate rocks can provide insights into local water column redox conditions, whereby negative Ce anomalies (values below 0.9–1), accompanied by seawater-like REE patterns, are commonly considered to indicate oxic conditions[20,21]. Samples with seawater-like REE patterns in this study show average Ce anomalies of 1.11 and 0.78 for the lower (0–32 m) and upper (50–120 m) parts of the succession, respectively. While we observed non-seawater-like REE patterns from the lower part (Supplementary Fig. 6; Supplementary Information), the exclusion of these values does not affect the overall Ce anomaly trend, which is consistent with previous studies[11,16] (Fig. 2). This indicates that the depositional settings transitioned from largely suboxic/anoxic seafloor conditions[20] during deposition of the lower part, to oxic conditions for the upper part. The very low sulfur contents in decarbonated residues of the samples[11] further indicate that the local seawater and/or pore water was mostly non-sulfidic during deposition of the lower part.

Carbonate-associated iodine, expressed as I/(Ca+Mg), is emerging as another useful proxy for tracking shallow seawater redox conditions[22]. This is mainly because only iodate is incorporated into carbonates[23], is prevalent in oxic seawater and reduces rapidly to iodide in low-oxygen environments[24]. Owing to the relatively slow oxidation kinetics of iodide[25,26], carbonate I/(Ca+Mg) ratios are also sensitive to oxycline proximity[27,28]. The low I/(Ca+Mg) values (< 0.5 μmol/mol) in our samples are consistent with most Proterozoic carbonate records (Fig. 2), which have been interpreted to indicate a redox-stratified water column with a shallow oxycline[29]. Nevertheless, I/(Ca+Mg) is highly susceptible to diagenesis[30] and generally decreases during diagenetic alteration[29]. Considering that the samples may have experienced neomorphism, diagenetic iodine loss cannot be ruled out entirely (Supplementary Information).

### Transient ocean deoxygenation
The sulfur isotope composition of carbonate-associated sulfate ($\delta^{34}S_{CAS}$) is a useful tool for reconstructing the dynamic sulfur cycle, which is linked to Earth's surface oxidation state through pyrite burial[31]. Recent advances in geochemical redox proxies have also allowed the extent of global seafloor anoxia to be estimated by uranium isotopes in carbonate rocks[16,32,33]. In short, during the reduction of soluble U(VI) to particulate-reactive U(IV), heavy $^{238}U$ is favoured due to the nuclear field shift effect[34]. Intensified U(IV) removal to sediments during periods of expanded marine anoxia thus drives seawater $\delta^{238}U$ toward lower values. Although there appears to be some effect from organic matter loading on uranium isotope fractionation[35], large isotopic fractionation is generally associated with uranium reduction. Therefore, marine anoxia is considered to exert a dominant control on the seawater $\delta^{238}U$.

Our carbonate $\delta^{238}U$ values (Fig. 2), combined with published $\delta^{238}U$ records[16] from different stratigraphic sections, reveal a decreasing trend following the Sturtian deglaciation (Fig. 3). The decreasing trend cannot readily be explained by sea-level changes or chemical weathering (Supplementary Information) and instead suggests significant expansion of anoxic waters. This evidence for expanded anoxia is accompanied by extremely high $\delta^{34}S_{CAS}$ values (mean 63‰; Supplementary Information), which require enhanced microbial sulfate reduction and removal of $^{32}S$-enriched pyrite from the ocean[36]. Therefore, the $\delta^{238}U$ and $\delta^{34}S_{CAS}$ data together suggest greatly expanded marine anoxia and pyrite burial following the Sturtian deglaciation. Indeed, coupled high $\delta^{34}S_{CAS}$ (> +40‰) and low $\delta^{238}U$ (<−1‰) values are also evident during the late Ediacaran[32,33,36,37] and similarly attributed to the spread of euxinic seawater and enhanced pyrite burial, along with an elevated isotopic ratio of sulfur input ($\delta^{34}S_{in}$)[33,36–38]. This coupling may also explain the observed trend following the Sturtian deglaciation, as glacial erosion could have exposed Tonian evaporites leading to elevated $\delta^{34}S_{in}$, while the expansion of euxinic seawater is further supported by iron speciation records[9] (Fig. 3).

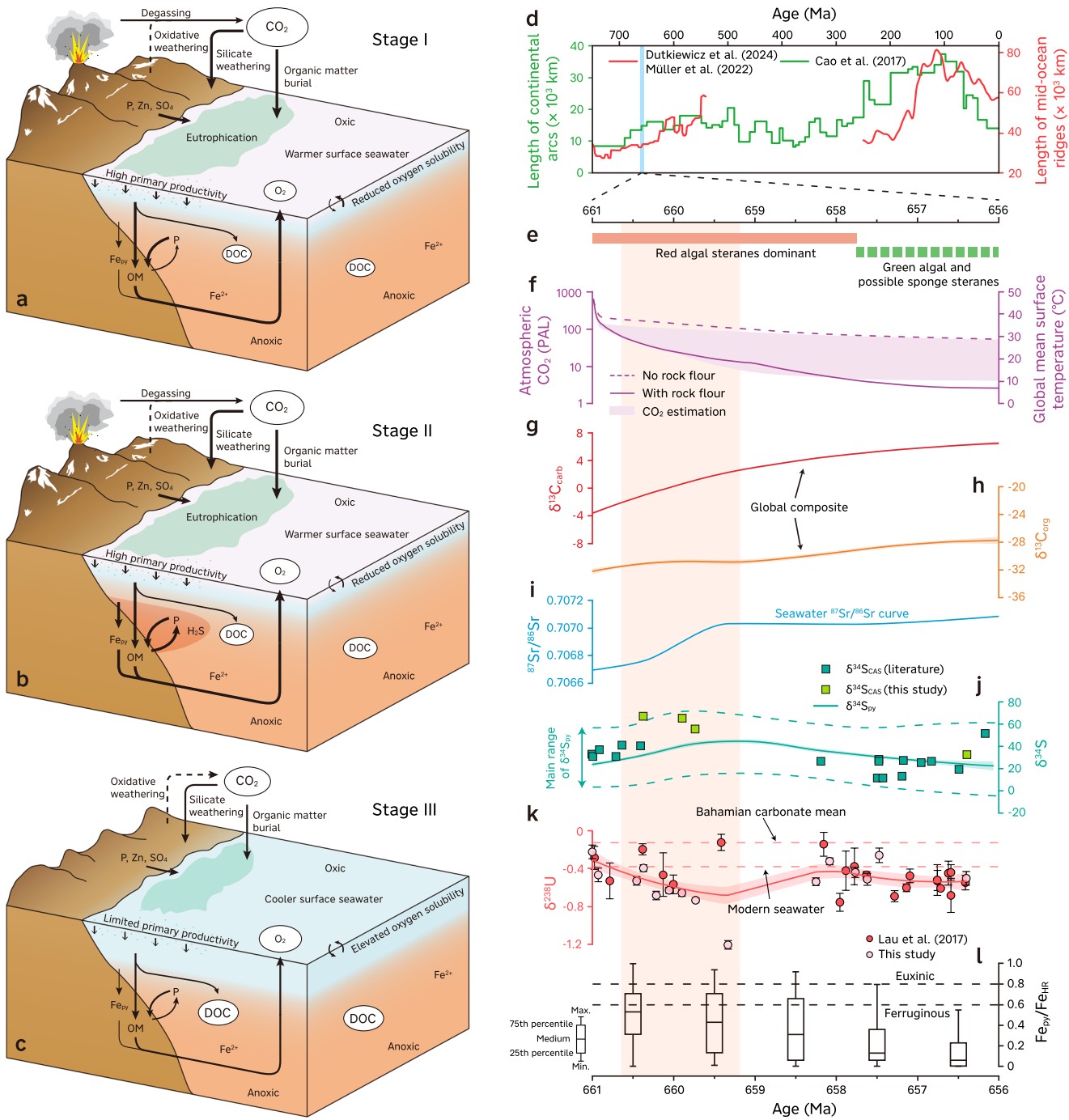

**Fig. 3 | Conceptual model and biogeochemical indicators following the Sturtian deglaciation. a** Vigorous weathering induced eutrophication after the Sturtian deglaciation. **b** Expansion of euxinia on productive margins owing to excess of $H_2S$ over iron. **c** Continuous atmospheric $CO_2$ drawdown led to subdued weathering, cooling and shallow seawater oxygenation. In (**a**–**c**), the thicknesses of arrows indicate the relative magnitude of fluxes, and the sizes of ellipses represent the relative sizes of reservoirs. **d** Global length of continental arcs[74] and mid-ocean ridges[75,101]. The blue band indicates the studied interval. **e** Occurrences of biomarkers[9]. **f** Possible evolution of atmospheric $CO_2$ and global mean surface temperature[9]. **g**–**l** Compilation of carbon, strontium, sulfur, uranium isotopes and iron speciation records (modified after ref. 9). Solid lines in (**g**, **h**, **j**, **k**) represent the LOESS (locally estimated scatterplot smoothing) fitting curves, and the shaded area represents 1σ (68%) confidence interval. The dashed lines in (**j**) indicate the major range of $\delta^{34}S_{py}$. In (**l**), each boxplot represents the distribution of global $Fe_{py}/Fe_{HR}$ data in each one-million-year time bin. The light shade of pale red band indicates the second stage. The age model of (**e**–**l**) is based on model A of ref. 9 (see Supplementary Fig. 2 for the alternative age model D). OM organic matter burial, $Fe_{py}$ pyrite burial, DOC dissolved organic carbon pool, PAL present atmospheric level, Min. minimum, Max. maximum.

The integrated carbonate $\delta^{238}U$ records then show a shift towards relatively higher values with a mean of −0.49‰ (Fig. 3). This $\delta^{238}U$ value has been associated with the post-Sturtian ocean oxygenation, with ~0.5% of the seafloor being anoxic compared to ~0.2% in the modern ocean[16,39]. Alternatively, this $\delta^{238}U$ value is similar to that of most mid-

Proterozoic carbonates, which has been interpreted to indicate limited euxinia in an ocean that was still largely anoxic[40]. Such a discrepancy is related to limited U isotope fractionation under suboxic and ferruginous conditions[41,42]. Recent studies suggest that U isotope fractionation associated with ferruginous conditions may vary significantly,

the magnitude of which is possibly linked to primary productivity with limited $\delta^{238}U$ fractionation under the low-productivity, ferruginous environment[43–45]. Given the potentially decreased primary productivity[46] (see also the following discussion), we consider that the positive $\delta^{238}U$ shift, instead of indicating widespread oxygenation, is more likely to reflect the return to less reducing conditions with a concomitant reduction in marine euxinia. The inference of a reduction in marine euxinia is in accordance with low $\delta^{98}Mo$ values in coeval shales[14,15] and decreasing $\delta^{34}S_{CAS}$ and pyrite $\delta^{34}S$ ($\delta^{34}S_{py}$) values (Fig. 3), which in turn implies an increase in sulfate levels[10]. Nevertheless, seawater sulfate concentrations overall remained reasonably low as evidenced by low $\Delta^{34}S$ ($\delta^{34}S_{CAS} - \delta^{34}S_{py}$) values[47,48] and low [CAS] in these samples (see Methods and Supplementary Data) and coeval carbonates[48]. Taken together, integrating $\delta^{238}U$ and $\delta^{34}S$ evidence suggests that the post-Sturtian ocean experienced a transient ocean deoxygenation marked by the expansion of low-sulfate euxinic seawater, followed by a return to less reducing conditions.

## Nutrient dynamics

Zinc is a bioessential micronutrient that functions as a cofactor in important enzymes such as carbonic anhydrase and alkaline phosphatase. The use of Zn isotopes to trace perturbations to micronutrient cycling is relatively new[49–52], but builds on extensive investigation of Zn isotope systematics. The modern deep ocean is mostly homogeneous with respect to $\delta^{66}Zn$ at around +0.45‰[53], which is isotopically heavier than riverine input (~0.33‰)[54,55]. Euxinic sediments could potentially record seawater $\delta^{66}Zn$ due to near quantitative removal of Zn[56,57]. Carbonate rocks are estimated to represent a minor sink for Zn[54,58]. Biogenic carbonates such as corals appear to be able to record the seawater $\delta^{66}Zn$ value[59], yet inorganic carbonates are isotopically heavier than the deep ocean with an offset of around 0.4‰ potentially due to $Zn^{2+}$ incorporation through tetrahedral coordination[58,60,61]. While Fe-Mn crusts and nodules are 0.4–0.5‰ heavier than seawater[54,55], recent studies suggest that the quantitatively most important oxic sink is dominated by pelagic oxic sediments[62], which is 0.1–0.2‰ lighter than seawater[58,63]. Importantly, the only significant sink of isotopically light Zn in largely anoxic oceans seems to be organic-rich continental margin sediments[64]. In light of this global Zn isotope mass balance framework, past variations in seawater Zn isotope compositions as recorded in carbonate rocks and/or shales have been used to infer changes in Zn input to, and output from, the ocean[49–52].

Carbonate $\delta^{66}Zn$ values exhibit a shift from +0.74‰ to +0.45‰, and remain around +0.45‰ in the lower part of the succession, followed by a gradual increase upsection to +0.68‰ (Fig. 2). The relatively low background $\delta^{66}Zn$ values compared to modern carbonates[61] might relate to the largely diminished pelagic oxic sink in deep ferruginous oceans. The $\delta^{66}Zn$ stratigraphic variation is unlikely to be accounted for by changes in relative sea-level and/or local redox conditions (Supplementary Information). Instead, considering the isotopic offset between carbonates and seawater, $\delta^{66}Zn$ variation in the lower part suggests a shift towards low regional seawater $\delta^{66}Zn$ values. This negative shift broadly coincides with the prominent rise of $^{87}Sr/^{86}Sr$ indicating significantly elevated weathering rates[11] (Fig. 2). Hence, one possible explanation is that strong postglacial weathering delivered abundant light Zn from continents either due to enhanced deglacial volcanism[10] and/or exposed large igneous provinces[9]. Shifts to lower seawater $\delta^{66}Zn$ values have also been attributed to a diminished light Zn sink (i.e., removal into organic-rich sediments) compensated for by expanded euxinic[50] or oxic (Fe-Mn crusts/nodules)[52] sinks, or to enhanced isotopically light Zn input from benthic[52,53] or hydrothermal[53] sources. A strong direct influence of hydrothermal Zn seems unlikely given the insignificant carbonate Eu anomalies in samples (~1.0), while increasing $^{87}Sr/^{86}Sr$ values are also inconsistent with a more globally significant hydrothermal source. Considering the U and S isotope evidence for expanding anoxia/euxinia during

deposition of the lower part, any increased oxic sink (Fe-Mn crusts/nodules) or benthic input due to oxygenation can also be largely excluded. An increased euxinic sink alone would lead to a decrease in the marine Zn reservoir and hence carbonate Zn concentrations[50], but Zn/(Ca+Mg) shows a consistent increasing trend in the lower part (Fig. 2). The relatively stable $\delta^{66}Zn$ during the expansion of anoxia/euxinia as indiated by $\delta^{238}U$ and $\delta^{34}S$ records (Fig. 2) also suggests that an expanded euxinic sink is unlikely to give rise to $\delta^{66}Zn$ variability in the lower part. On the other hand, the $\delta^{66}Zn$ values then gradually increase upsection, and its onset occurs prior to the shift to less reducing conditions as indicated by Ce/Ce* and U isotopes (Fig. 2), implying a limited role for an expanding oxic sink. Furthermore, the transition to less reducing conditions in the upper part may have decreased seawater $\delta^{66}Zn$ values via remobilization of light Zn from continental margin sediments[52]. Considering the coincidence with the plateauing of $^{87}Sr/^{86}Sr$ ratios, the increase in $\delta^{66}Zn$ suggests that weathering-derived light Zn input was progressively outweighed by a relative increase in the burial of isotopically light organic-rich sediments. Overall, $\delta^{66}Zn$ evidence suggests an enhanced nutrient supply into the ocean following the Sturtian deglaciation likely leading to eutrophication, followed by a period of diminished influx.

Phosphorus is considered to be the ultimate limiting nutrient for primary productivity in oceans over geological timescales[65]. Carbonate P/(Ca+Mg) can record relative changes to oceanic phosphate levels in deep time as long as other potential influences can be deconvolved[66,67]. The lower part of the succession has a relatively higher mean P/(Ca+Mg) (0.11 mmol/mol) ratio with a prominent peak to 0.2 mmol/mol near its top, compared to the upper part (mean 0.08 mmol/mol) (Fig. 2). The partitioning of phosphate into carbonate minerals is affected by the fluid phosphate concentration, pH, temperature, alkalinity, mineralogy and precipitation rates[68]. Following the post-Sturtian super-greenhouse climate, the oceans could have received enhanced alkalinity and calcium input from weathering[13], which might have increased carbonate precipitation rates[68]. Meanwhile, there seem to be no significant variations in ocean pH[69] and mineralogy (Supplementary Information). The combined effects would likely decrease carbonate P/(Ca+Mg) values, and hence, the prominent peak in the lower part implies relatively elevated seawater phosphate levels. Because of the nutrient-type distribution of dissolved phosphate in the ocean, shallow seawater would be expected to exhibit phosphate depletion, making the prominent increase with basin shallowing even more remarkable. The peak of P/(Ca+Mg) broadly coincides with high $\delta^{34}S_{CAS}$ and low $\delta^{238}U$ values, suggesting that the increase in dissolved phosphate concentrations resulted from low P burial efficiency due to intensified recycling under euxinic/sulfidic conditions[70]. This positive nutrient recycling feedback is also supported by phosphorus phase association and iron speciation records from coeval strata[9].

## Weathering induced biogeochemical cascades

Redox dynamics and the global carbon cycle following the Sturtian deglaciation were dictated by an extreme climatic, but also a highly unusual tectonic regime (Fig. 3). Specifically, the peneplanation of Rodinia after its amalgamation[71] likely led to a relatively low-relief supercontinent, on which chemical weathering was more transport- than rate-limited. During the long-lasting Sturtian glaciation, ice sheets scoured these relatively flat-lying continental interiors, removing regolith and surface rocks, and producing highly reactive rock flour[6,72]. As a consequence, the high $pCO_2$ levels accumulated during glaciation decreased rapidly after deglaciation due to enhanced silicate weathering[13,72] but were not readily replenished due to unusually low arc volcanism[73,74] and mid-ocean ridge carbon outflux[75] (Fig. 3). The brief episode of intense chemical weathering that followed deglaciation led therefore into a more prolonged interval of global cooling, in accordance with the temporal evolution of seawater $^{87}Sr/^{86}Sr$ (Fig. 3), and markedly lower nutrient and sulfate delivery to the ocean[46,72].

The ocean at the end of the Sturtian deglaciation (~661 Ma) was probably dominated by anoxic ferruginous conditions (Fig. 3a) with euxinia delayed by limited sulfate availability in the oceans and a high proportional influx of iron relative to sulfate[6]. With increasing nutrient and sulfate input, a tipping point must have been crossed whereby a relative excess of $H_2S$ over iron promoted the expansion of euxinia. Enhanced P recycling under euxinic/sulfidic conditions in turn exerted a positive productivity feedback[9,70] (Fig. 3b). Consequently, elevated organic carbon and pyrite burial, as evidenced by a coupled increase in $\delta^{13}C_{carb}$ and $\delta^{34}S$, would have resulted in a pulse of oxygen production. However, stimulated primary production would also have increased organic export to the deep ocean, increasing the oxygen demand for organic matter remineralisation[76]. The overall low seawater sulfate levels may also have increased the seabed methane flux, helping to keep seafloors anoxic[77]. Hence, increased oxygen consumption in the marine realm associated with organic matter (and possibly methane) oxidation resulted in ocean deoxygenation.

Following the eutrophication episode, the gradual decrease in weathering-derived nutrient and sulfate influx led to a lessening of marine euxinia[9], in part due to a negative productivity feedback caused by less efficient P recycling (Fig. 3c). Together with increased oxygen solubility driven by a cooling climate[78], this change would have been conducive to the expansion of less reducing environments in the marine realm. The cooling climate could also have decreased microbial respiration and hence induced a more efficient biological carbon pump[79]. In the absence of abundant oxidants, this process may have favoured organic matter preservation as supported by increasing $\delta^{13}C_{carb}$ (Fig. 3). Nonetheless, $\delta^{34}S$ records broadly show an inverse relationship with $\delta^{13}C_{carb}$ during this stage (Fig. 3). Such a decoupling appears to suggest that the oxidant (oxygen) generated by organic carbon burial was partly transferred into sulfate through a surplus of pyrite weathering over burial, thereby limiting the extent of atmospheric oxygenation. Collectively, the above scenarios illustrate that oxygenation of the atmosphere and oceans (especially the deep marine realm) was likely asynchronous during the Cryogenian nonglacial interlude.

### Implications for the carbon cycle and biological evolution

Sustained high $\delta^{13}C_{carb}$ values during the Cryogenian nonglacial interlude are conventionally interpreted to infer elevated organic carbon burial rates[8]. We suggest that the redox and nutrient dynamics, as discussed earlier, should also have enlarged the marine DOC inventory and hence impacted the carbon cycle. During post-glacial eutrophication, abundant organic substrates were available but would not have been efficiently degraded under expanded anoxia conditions, thus promoting DOC accumulation[80]. The later nutrient-limited condition, in conjunction with decreased microbial respiration driven by a cooling climate, would also have limited microbial DOC consumption[81–83]. Therefore, post-Sturtian environmental change led to the net accumulation of a recalcitrant DOC pool in the deep ocean, which is supported by a relatively muted $\delta^{13}C_{org}$ shift compared with the prominent rise of $\delta^{13}C_{carb}$ (Fig. 3). The DOC pool could have switched from net accumulation to net oxidation once certain oxidant thresholds were crossed, buffering against extensive oceanic oxygenation and atmospheric $CO_2$ drawdown[78,84]. Such a prediction is consistent with coincident negative shifts of both $\delta^{238}U$ and $\delta^{13}C_{carb}$ through the Taishir anomaly[16]. Taken together, we suggest that the sustained high $\delta^{13}C_{carb}$ following the Sturtian Snowball deglaciation is at least in part due to the buildup of a deep ocean DOC pool.

Marine anoxia and high sea surface temperatures have been invoked to explain the delayed rise of green algae and putative demosponges after the Sturtian deglaciation[5,9,10]. There is a broad coincidence between the development of less reducing environments, cooling climate and the rise of biotic complexity (Fig. 3). A synergistic effect can be envisioned whereby a lessening of marine anoxia alongside limited nutrient availability and moderate climate would have extended the habitable space, perhaps setting the stage for the expansion of opportunistic species. Therefore, our dataset appears to support the idea that severe environmental stress after the Sturtian deglaciation delayed further biological diversification. Nonetheless, conditions would have been favourable only to aerobic forms with low oxygen demands because the oceans were still not significantly oxygenated. Later oceans remained largely anoxic and the episodic transitions towards a more oxygenated system throughout the late Neoproterozoic appear to have relied on external stimuli such as tectonic drivers. This reliance may explain the prolonged rise of oxygen and delayed occurrence of energetic lifestyles over the late Ediacaran to Cambrian.

## Methods

Bulk limestone samples were crushed into small rock chips. The clean chips without visible veins and weathered surfaces were selected and ground into fine powders using a TEMA agate disc mill or agate mortar and pestle for further processing.

### Rare earth elements and phosphorus

Rare earth elements (REE) and phosphorus were extracted using sequential leaching[85]. For each sample, ~50 mg powder was prewashed with 5 ml neutral 1 M ammonium acetate. The mixture was sonicated in the ultrasonic bath for 30 min and then centrifuged for 5 min, from which the supernatants were discarded. The residues were rinsed with de-ionized (DI) water (18.2 MΩ·cm) three times, followed by partial leaching with dilute acetic acid (0.3 M) for 30 min in the ultrasonic bath. The resultant supernatants obtained by centrifugation were filtered into acid cleaned Teflon beakers using 0.22 μm syringe filters, dried on a hot plate and redissolved in 2% $HNO_3$. The solutions were further diluted in 2% $HNO_3$ for major and minor element analysis (Ca, Mg, Fe, Mn, Al, P, Sr, Ba) on a Varian 720 inductively coupled plasma optical emission spectrometer (ICP-OES) at University College London. Phosphorus was analysed at a wavelength of 213.618 nm using the polyboost function. The leachates were then diluted to ~100 ppm Ca with 2% $HNO_3$ for rare earth elements analysis via an Agilent 7900 inductively coupled plasma mass spectrometer (ICP-MS) at University College London. The internal standard of 2 ppb indium was added to all analysed solutions to monitor the instrumental drift and matrix effects. The formation of 2+ ions and oxide interference were monitored using $Ba^{2+}$ and the formation rate of Ce oxide before analysis. Correction of BaO interference on Eu was achieved by monitoring single Ba elemental solutions throughout the analytical session. The accuracy was monitored by two in-house bulk digested solution standards of SRM-88a and SGR-1, which showed agreeable values with certified values within uncertainty. Repeated analysis of the drift monitor and standards gave a precision better than 5%. Blanks showed negligible concentrations below the detection limit. REE concentrations are normalised against post-Archean Australian Shale[86] (denoted by the subscript N). The bell-shaped index (BSI) is calculated after ref. 87 to evaluate the enrichment of middle REE. The Ce anomaly is calculated following ref. 88 $Ce/Ce^* = Ce_N \times Nd_N/Pr_N^2$.

### Carbonate-associated iodine

The carbonate-associated iodine (CAI) was extracted according to an updated method[89]. Around 10 mg powder was sonicated in 1 ml DI water that was then centrifuged and discarded. The residue was leached with 1 ml 3% $HNO_3$ in the ultrasonic bath for 10 min. After centrifugation, an aliquot of the supernatant was pipetted into the stabiliser (3% ethylenediaminetetraacetic acid in 3% ammonium hydroxide). Such stock solution was then diluted in the stabiliser for Ca and Mg analysis using a Varian 720 ICP-OES at University College London. The accuracy was checked with two in-house solution standards of SRM-88a and SGR-1, and blanks were negligible. Based on the results, the stock solution was further diluted to ~100 ppm Ca with the stabiliser for iodine analysis on an Agilent 7900 ICP-MS at University

College London. Calibration standards were made freshly from potassium iodate powder, dissolved in the stabiliser with 100 ppm Ca matched to samples. The detection limit of I/Ca is -0.1 µmol/mol. Repeated analysis of the drift monitor gave the relative standard deviation better than 3%. Analyses of JCp-1 and CRM-393 duplicates yielded I/Ca values of 3.90 ± 0.04 µmol/mol (1 SD, $n = 3$) and 0.39 µmol/mol ± 0.001 (1 SD, $n = 3$), which were comparable to published literature values[22,89]. The stabilisers and procedure blanks showed negligible iodine concentrations.

## Carbonate-associated sulfate

Approximately 15–30 g of sample powder was first immersed in excess 12% NaClO followed by ultrasonication in a water bath for 1 h and shaken overnight. The residues after centrifugation were rinsed with DI water three times. The bleaching process was repeated five times, after which the residues were saturated in excess 1% $H_2O_2$ overnight under constant agitation. The residues were rinsed with DI water three times and dried out in the oven. The dried residues were cleaned with excess 10% NaCl for 24 h on an orbital shaker and then rinsed with DI water three times. The cleaning process was repeated five times, after which the residues were dissolved in 6 M HCl within 45 min. The leachates were filtered through 0.22 µm syringe filters immediately, to which a 200 g/L $BaCl_2$ solution was added to precipitate $BaSO_4$. The resultant precipitates were washed with 6 M HCl once and DI water three times before being dried and weighed for sulfur isotope analysis. This multistep process for carbonate-associated sulfate (CAS) extraction follows the method of ref. 90 to minimise any potential contamination from non-carbonate phase sulfur. No precipitate was observed from the blank. Sulfur isotope analysis of dried precipitates ($\delta^{34}S_{CAS}$) was carried out at Iso-Analytical Laboratory (UK) using the EA-IRMS. The sample and vanadium pentoxide catalyst contained in a tin capsule were combusted at 1080 °C in the presence of oxygen, which was further combusted by passing through tungstic oxide/zirconium oxide in a helium stream, followed by reduction by high purity copper wires. Water is removed using a Nafion™ membrane, while $SO_2$ is resolved from $N_2$ and $CO_2$ on a packed gas chromatograph column at a temperature of 45 °C before it was introduced into the IRMS. The analysis was based on monitoring of $m/z$ 48, 49 and 50 of $SO^+$ produced from $SO_2$, and $\delta^{34}S$ values are reported relative to the Vienna Canon Diablo Troilite (VCDT) standard. Three lab working standards (IA-R025, IA-R026, IA-R061) were used for calibration, which have been analysed against the international standards NBS-127 (+20.3‰), IAEA-SO-5 (+0.5‰) and IAEA-S-1 (−0.3‰). The accuracy was monitored by analysing IA-R061 (+20.51‰ ± 0.11, 1 s.d., $n = 3$) and IAEA-SO-5 (+0.71‰ ± 0.03, 1 s.d., $n = 2$), which are consistent with reported values. The CAS concentrations were calculated based on the weights of dried precipitates and the sample purity determined during sulfur isotope analysis. It should be noted that while we processed all the samples, only five samples yield sufficient sulfur for isotopic analyses.

## Uranium isotopes

Approximately 500 mg sample powder was partially leached with 20 ml 0.2 M acetic acid for 24 h at room temperature in order to limit the contamination of uranium associated with non-carbonate phases[91]. Following centrifugation, the supernatants were filtered through 0.22 µm syringe filters, aliquoted and diluted for measurement of elemental concentrations (Ca, Mg, Fe, Mn, Al, Sr, U) using ICP-OES and ICP-MS at University College London. The accuracy was assessed by the analysis of in-house solution standards of TMDA-70, SGR-1, and SRM-88a, which were in agreement with certified values. Repeated analysis of the drift monitor gave the relative standard deviation better than 5%. Based on measured U concentrations, around 40 ng U was pipetted into acid-cleaned Teflon beakers and mixed with IRMM 3636 uranium double spike to achieve a spike/sample ratio of approximately 0.1. The mixtures were dried down and

redissolved in 1 ml 3 M $HNO_3$. Uranium was then purified using Eichrom UTEVA resin following the protocol of ref. 92. Briefly, columns and resin were cleaned with 0.05 M HCl and conditioned with 3 M $HNO_3$ before loading samples. The matrix elements and Th were eluted using 3 M $HNO_3$ and 6 M HCl, respectively, and U was collected with 0.05 M HCl. Purified samples were oxidised with concentrated $HNO_3$ and redissolved in 3% $HNO_3$ before analysis at Royal Holloway University of London. Uranium isotope ratios were measured on a Thermo-Finnigan Neptune Plus multi-collector inductively coupled plasma mass spectrometer (MC-ICP-MS) equipped with an Aridus III desolvating nebulizer system for sample introduction. Acid blank was analysed before every solution analysis to correct measured intensities. Uranium isotope ratios are reported relative to CRM-112A standard:

$$\delta^{238}U = \left[ \left( {}^{238}U/{}^{235}U_{sample} \right) / \left( {}^{238}U/{}^{235}U_{CRM-112A} \right) - 1 \right] \times 1000$$

Propagated analytical uncertainties for individual $\delta^{238}U$ measurements were typically 0.03–0.08‰ (2 SE) for -15 ppb solutions. A limestone standard CRM-393 was processed in the same manner as samples and gave a $\delta^{238}U$ of -0.35 ± 0.08‰ (2 SE), which agrees with known values. Two duplicates were processed through the same full procedure and showed excellent agreement. Total procedural blank was -5 pg of U and negligible. The error is reported as propagated uncertainty on individual isotope ratios.

## Zinc isotopes

Carbonate-bound zinc was extracted following the protocol of ref. 61. Around 200 mg of sample powder was rinsed with DI water three times, then leached with 20 ml buffered 1 M ammonium acetate solution (pH 5) for 24 h at room temperature under constant agitation. The supernatants after centrifugation were pipetted into new acid-cleaned tubes. The aliquots of leachates were diluted in 2% $HNO_3$ for elemental analysis (Ca, Mg, Fe, Mn, Al, Sr, Zn) using ICP-OES and ICP-MS at University College London. The in-house solution standards of TMDA-70, SGR-1, and SRM-88a were analysed to check the accuracy, which exhibited values consistent with certified values. Repeated analysis of the drift monitor gave the relative standard deviation better than 5%. Based on the Zn concentrations, 100–200 ng Zn was pipetted into acid-cleaned Teflon beakers, which were dried down on the hotplate, oxidised with excess concentrated $HNO_3$ and refluxed with 1 ml 6 M HCl. The certain amount of a ${}^{64}Zn$-${}^{67}Zn$ double spike was added to all samples to achieve a spike/sample ratio of approximately -1.1–1.2[93], followed by evaporation to dryness and refluxed with 1 ml 6 M HCl. Solutions were further dried down and redissolved in 1 ml 1 M HCl to ensure spike-sample equilibration. Zinc was then purified through column chromatography using AG MP-1M resin (BioRad) following refs. 54,94. In brief, columns and resin were cleaned with 2% $HNO_3$ and conditioned with 1 M HCl before loading samples. The matrix elements were eluted with 1 M HCl, and Zn was collected with 0.01 M HCl. Prior to analysis, purified samples were oxidised with concentrated $HNO_3$ and redissolved in 2% $HNO_3$. Zinc isotopes were measured on a Nu Instruments Plasma 3 MC-ICP-MS at University College London following the protocol of refs. 59,93,95. Solutions of -100 ppb Zn were introduced via an Aridus III desolvating nebulizer system and a PFA nebulizer (50 µL/min). A standard-sample bracketing approach was adopted during analysis, and the raw data collection was preceded by an analysis of 2% $HNO_3$ for signal correction. Interference corrections for ${}^{64}Ni$ and $Ba^{2+}$ were negligible. Instrumental mass bias was corrected through the double-spike technique[93] using the off-line data reduction procedure of ref. 96. Zinc isotope ratios were determined relative to a new standard AA-ETH Zn:

$$\delta^{66}Zn = \left[ \left( {}^{66}Zn/{}^{64}Zn_{sample} \right) / \left( {}^{66}Zn/{}^{64}Zn_{AA-ETH} \right) - 1 \right] \times 1000$$

Final $\delta^{66}$Zn values are reported with respect to JMC-Lyon by applying a correction of +0.28‰[97]. Total procedural blanks were ~1–2 ng. Accuracy and external reproducibility of $\delta^{66}$Zn values were assessed by two secondary solution standards (London Zn and NCS DC70303) and a limestone standard (CRM-393) that was processed through the same full procedure as the samples. Solution standards gave a $\delta^{66}$Zn of 0.11 ± 0.04‰ (2 SD, $n = 14$) for London Zn and a $\delta^{66}$Zn of 0.73 ± 0.06‰ (2 SD, $n = 6$) for NCS DC70303, which are consistent with previously reported values[59]. CRM-393 gave a $\delta^{66}$Zn of 0.75 ± 0.05‰ (2 SD, $n = 4$). Three duplicates were processed along with samples to assess internal reproducibility and yielded agreeable values within the error of uncertainty. The reported error on figures is the external 2 SD reproducibility of NCS DC70303, which was the largest external uncertainty during the measurement period.

## Ethics and inclusion
We affirm that all geological materials were collected in a responsible manner and in accordance with relevant permits and local laws. Local and regional research relevant to this study has been cited where appropriate.

## Data availability
The original data generated in this study are provided in the Supplementary Information and deposited in the Figshare repository (https://doi.org/10.6084/m9.figshare.29135669).

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

## Acknowledgements

We would like to thank E.E. Stüeken and Xi Chen for insightful discussions; Zheyu Tian and Lin Yuan for help with experiments; A. Basu and G. Tarbuck for technical support. K.Z. was financially supported by the China Scholarship Council, University College London Faculty Dean's Prize and International Association of Sedimentologists (IAS) Postgraduate Research Grant. G.S. acknowledges funding support from the joint NERC-NSFC Biosphere Evolution Transitions and Resilience (BETR) programme NE/P013643/1 and NERC project NE/R010129/1. G.S. gratefully acknowledges the financial support of the John Templeton Foundation (#62220). The opinions expressed in this paper are those of the authors and not those of the John Templeton Foundation.

## Author contributions

G.A.S. and K.Z. conceived the study; G.A.S. supervised the project; S.H.L., A.J.D. and K.Z. contributed to the generation of geochemical data; K.Z. wrote the initial draft of the manuscript. All authors contributed to reviewing and editing the manuscript at all stages.

## Competing interests
The authors declare no competing interests.
