## [Transparent Peer Review file · Nature Communications]

Ocean deoxygenation after the Sturtian Snowball

Corresponding Author: Dr Kun Zhang

Version 0:

Reviewer comments:

Reviewer #1

(Remarks to the Author)

Zhang and co-authors present a suite of novel paleoenvironmental and paleonutrient proxies (CAS S isotopes, CAU isotopes, iodine ratios, REE patterns and Ce anomaly, Zn isotopes, and CAP) in a carbonate succession from Mongolia deposited after the Sturtian glaciation. They compare their data to published records and present a model for variable redox conditions, nutrient delivery, and climate events to explain the geochemical records. They also speculate that this combination of biogeochemical conditions could explain the anomalous C isotope behavior observed during this time. This manuscript presents a large amount of novel data that are rarely discussed in tandem and therefore this multi-proxy dataset is notable. It is not a simple task to discuss a diversity of proxy data in a digestible way and the authors have done a nice job in this regard. The time interval is also of broad interest for its geobiological importance. Especially given the wealth of data, the paper is logically structured and the authors consider many different angles (contamination, diagenesis, sea level etc.) in the supplement. The interpretation is mostly grounded in the data in this paper and presented elsewhere.

1. My main comment is that there is a lot of attention and interpretation on the basal 10's of meters of the section (the lowermost ~30 m in Figure 2 and stages I and II in Figure 3) that are interpreted to represent a period of transient deoxygenation/euxinia and intense eutrophication. This would be novel because it resolves some of the differences from previous studies. However, the geochemistry of these strata should be interpreted with caution and I encourage the authors to consider alternative processes driving this portion of the data. The immediate deglacial cap carbonate and postglacial sequence have been proposed to represent a poorly mixed ocean resulting from a fresh meltwater lens that was strongly stratified from saline ocean water that evolved under snowball conditions (e.g., Yang et al., 2017 *Geology*; Hoffmann et al. 2017 *Science Advances* p.26-31). The mixing time scale was modeled to be 10^4 to 10^5 yr (Yang et al., 2017), and the unusual ocean dynamics of the deglacial and postglacial period present a challenge in interpreting the basal samples as representative of a global seawater signal. The duration of a stable, density-stratified ocean is also difficult to determine and likely also is variable depending on the specific oceanography of a given site, but caution in interpreting geochemical data in the "T1" member of the Taishir Formation is warranted (Lau et al., 2017).

The potential for a stably stratified ocean to influence carbonates in the Taishir Formation is supported by data presented in this study. Specifically, the REE patterns in Figure S4, which are discussed in the supplement, lack features diagnostic of seawater (heavy REE enrichment compared to light REE enrichment, inconsistently or lack of a developed Y anomaly, relatively low Pr and Nd and relatively high middle REEs). Lines 132-134 of the supplement attribute the "non-seawater-like REE patterns" to early diagenesis but it is also reasonable to consider that these are non-seawater-like because they are in fact recording a syndepositional signal of poorly mixed fluids. Attributing these patterns to early diagenesis is also contradictory to the supplemental discussion that these samples do not otherwise correlate with other geochemical tests or markers for diagenesis. Therefore, this alternative explanation for the REE patterns (as well as the fluid influencing these carbonates) is feasible and should be considered. Given this prior work, the authors should consider the possibility that the basal ~32 m of section should not be interpreted in the context of global secular seawater evolution. If this were to be the case, what are the impacts on the overall interpretation of environmental change after the Sturtian snowball?

2. Carbonate U isotope records. The interpretation of the U isotopes is primarily driven by one quite negative $\delta^{238}\text{U}$ value (of ~-1.2‰ at ~32 m). This value is low for carbonates, for this section, and for this time. In addition to the lower samples not containing a seawater-like REE sample and for the potential for a poorly mixed, stratified ocean to be driving the geochemistry of the basal "Sequence 3", there is also potential for this sample to be an outlier. The supplement discusses that this sample has anomalously high Al and dismisses siliciclastic contamination; I think this reasoning is solid, but it still remains that there is something unusual about this sample. It is possible that this sample is representative of a larger trend,

but without further samples from this section it is not possible to interpret this confidently. Moreover, Figure 3 compares the Tsagaan Gorge U isotope record with that from other carbonate sections from this basin in Lau et al. (2017). When combined, the significance of this one sample as well as the proposed negative trend in $d_{238}\text{U}$ is less obvious. Why not combine the entire record if the U isotope record in the interpretation if a compilation of global S isotope data are also interpreted in reconstructing redox conditions? How would a broader compilation of data from the Taishir Formation modify the interpretations of anoxia  to a more oxidizing ocean?

3. Iodine to calcium+magnesium ratios. Iodine ratios are particularly susceptible to diagenesis and will be readily altered before other carbonate geochemical proxies (see Hardisty et al., 2017, cited in the main text; Lau and Hardisty, 2022 GCA). Importantly, iodine ratios that are below detection are also the result of diagenetic alteration. Therefore, the interpretations of iodine ratios below detection as anoxic (or positioned near the oxycline) are possible but cannot be done with confidence when the alternative of diagenetic alteration is also possible. Using the positive Ce anomaly of the lowermost strata to support the interpretations of anoxia isn't convincing because both myself and the authors state that the REE patterns are not representative of seawater. In addition, low $I/(\text{Ca}+\text{Mg})$ may also represent a redox stratified ocean and not just local anoxia. The reason for this is because the kinetics of iodide to iodate oxidation are slow, so an ocean poorly oxygenated bottom waters will not be able to produce a large iodate reservoir (Hardisty et al., 2021 EPSL).

3. Age model and correlation of the Taishir Formation to the global compilation. The overall interpretation of the new geochemical data presented here (Figure 3) relies on comparison to the global S isotope compilation, the Fe speciation compilation, the biomarker compilation, and modeled climate data--none of which are from Mongolia. It should be noted that the age model discussion in the supplement proposes that the time interval covered by the studied section is 2-4 Myr (line 83), which is well shorter than the six million years shown in Figure 3.

Comparing records from different sites (e.g., Mongolia to other sites) should be done cautiously as it relies on a highly speculative correlation and age model. The authors acknowledge the challenge here in the supplement, and for correlating Mongolia specifically (lines 68-85). The timing of deglaciation is potentially diachronous (Rooney et al., 2020 Geology) and the duration of the carbonate units within the interglacial period are very difficult to constrain. The age model proposed by Bowyer et al. and presented in Figure 3 proposes that the duration of the T1 member is ~two million years. Based on C isotope chemostratigraphy of a unit that may not be representative of seawater, the age here is not well constrained given the thickness of the unit relative to T2 and T3 and the potential for the cap carbonate at the base of T1 to represent an unusually high sedimentation rate. The challenge of correlating Cryogenian units is certainly not unique to this study. However, because the interpretations combine S isotopes from elsewhere and the implications of this study are interpreted with other geochemical, biomarker, climate, and tectonic data collected elsewhere, additional nuance should be added to the conceptual model to account for this uncertainty.

Minor comments

Line 61: "high purity samples" – what does this mean?

As a note, the authors discuss that previous interpretation of the U isotope record grouped together the ferruginous and euxinic sinks (Lau et al., 2017). Others have proposed that if suboxic or ferruginous sediments have a more muted isotope fractionation compared to sulfidic sediments, the relatively high $d_{238}\text{U}$ in the lower part of the Taishir Formation (compared to the T3 member which is not discussed here) could represent not oxic but suboxic or ferruginous conditions, at the expense of sulfidic sediments. It should be noted that evidence for ferruginous and euxinic conditions having distinct U isotope fractionation is equivocal (e.g., Lau et al., 2022 GCA, Rutledge et al., 2024 GCA). Nonetheless, I do want to emphasize that I am excited that, while this study interprets the extent of oxygenation to be less widespread than in the 2017 paper, the overall trend in less reducing conditions in T2 compared to T3 is also shown here, and the data in T2 are generally well aligned. (Note that T1 was interpreted with caution due to the potential for deglacial and postglacial stratification leading to non-steady-state behavior and large scatter).

Figure 3

- Panel J: it is hard to distinguish between literature data and data presented in this study. The colors should be changed.
- Yellow bar is also difficult to see (on both printed PDF and on my screen)

- Kimberly Lau, Penn State

Reviewer #2

(Remarks to the Author)

In this manuscript, Zhang et al. present a new multi-proxy carbonate dataset from Mongolia that spans the interval immediately following the Sturtian glaciation. They argue for an initial pulse of eutrophication and widespread anoxia, followed by a return to less reducing but not fully oxygenated ocean conditions. Overall, I find the manuscript to be comprehensive and well-written, and the interpretations are broadly supported by the data. I recommend publication with minor to moderate revisions. I have given detailed line-by-line comments in both the main text and the supplement (attached files), so please follow those closely. Overall, I feel that some more explanations are needed about some specific diagenetic issues such as fractionation of Zn isotopes during incorporation into carbonate and the correlations between $d_{34}\text{SCAS}$ and variables such as $[\text{Al}]$ and Sr/Ca . Including a plate of photomicrographs in the supplement would be helpful to the reader. I point out numerous issues such as this in my comments, but overall I am positive about this manuscript and don't have any

overarching criticisms that should prevent publication.

Version 1:

Reviewer comments:

Reviewer #1

(Remarks to the Author)

I reviewed the initial submitted manuscript. I appreciate the consideration of my previous comments and concerns, which the authors have done satisfactorily. Specifically, the new discussion in the supplement and in the rebuttal addresses my main comment about the meltwater pulse and I appreciate that the revised manuscript has integrated the data more completely than in the initial version.

I have some minor comments/suggestions prior to acceptance.

Lines 172-175: This is an interesting hypothesis which is made compelling by the tie-in to the productivity trends.

Line 179: "in the study samples" or "in these samples"

Line 216: Given that burial of organic matter can result from multiple factors, I suggest connecting organic-rich continental margin sediments to organic carbon flux to productivity more explicitly here.

Lines 244-246: can the link between expanded euxinia and $\delta^{66}\text{Zn}$ be made more clear? I didn't quite follow this.

Lines 247-249: Perhaps, but what if it's local redox change and not global redox change that is driving the Zn isotope record? It seems like (also?) pulling in the Ce/Ce* would be useful in making this argument

Line 253: This paragraph discusses several mechanisms and trends in Zn isotopes. It is useful but I wasn't clear what the sum of the arguments boiled down to with regard to what the authors are interpreting from the Zn record. It would be useful to add a summary sentence that links the record back to what can be confidently stated about nutrient cycling/availability. i.e. extend the statement about burial flux of isotopically light organic rich sediments outpacing weathering to when and how productivity increased.

Fig 2. Now some vertical lines are not labeled/explained in caption. I think this was useful in the previous version, even if there was some debate over which threshold was meaningful. That could be explained further in the caption.

Fig S2 replicates Fig 3?

Reviewer #2

(Remarks to the Author)

The authors have done a nice job revising the manuscript. I do have additional comments that should be addressed in the attached annotated PDF, but they are now relatively minor. Note the comments are in the merged PDF in the portion with the accepted (not tracked) new manuscript. I recommend publication once these few remaining issues are addressed.

Response to Reviewer Comments

Reviewer #1

Zhang and co-authors present a suite of novel paleoenvironmental and paleonutrient proxies (CAS S isotopes, CAU isotopes, iodine ratios, REE patterns and Ce anomaly, Zn isotopes, and CAP) in a carbonate succession from Mongolia deposited after the Sturtian glaciation. They compare their data to published records and present a model for variable redox conditions, nutrient delivery, and climate events to explain the geochemical records. They also speculate that this combination of biogeochemical conditions could explain the anomalous C isotope behavior observed during this time. This manuscript presents a large amount of novel data that are rarely discussed in tandem and therefore this multi-proxy dataset is notable. It is not a simple task to discuss a diversity of proxy data in a digestible way and the authors have done a nice job in this regard. The time interval is also of broad interest for its geobiological importance. Especially given the wealth of data, the paper is logically structured and the authors consider many different angles (contamination, diagenesis, sea level etc.) in the supplement. The interpretation is mostly grounded in the data in this paper and presented elsewhere.

We thank Kimberly Lau for the constructive comments on our manuscript.

1. My main comment is that there is a lot of attention and interpretation on the basal 10's of meters of the section (the lowermost ~30 m in Figure 2 and stages I and II in Figure 3) that are interpreted to represent a period of transient deoxygenation/euxinia and intense eutrophication. This would be novel because it resolves some of the differences from previous studies. However, the geochemistry of these strata should be interpreted with caution and I encourage the authors to consider alternative processes driving this portion of the data. The immediate deglacial cap carbonate and postglacial sequence have been proposed to represent a poorly mixed ocean resulting from a fresh meltwater lens that was strongly stratified from saline ocean water that evolved under snowball conditions (e.g., Yang et al., 2017 *Geology*; Hoffmann et al. 2017 *Science Advances* p.26-31). The mixing time scale

was modeled to be 10^4 to 10^5 yr (Yang et al., 2017), and the unusual ocean dynamics of the deglacial and postglacial period present a challenge in interpreting the basal samples as representative of a global seawater signal. The duration of a stable, density-stratified ocean is also difficult to determine and likely also is variable depending on the specific oceanography of a given site, but caution in interpreting geochemical data in the “T1” member of the Taishir Formation is warranted (Lau et al., 2017).

The potential for a stably stratified ocean to influence carbonates in the Taishir Formation is supported by data presented in this study. Specifically, the REE patterns in Figure S4, which are discussed in the supplement, lack features diagnostic of seawater (heavy REE enrichment compared to light REE enrichment, inconsistently or lack of a developed Y anomaly, relatively low Pr and Nd and relatively high middle REEs). Lines 132-134 of the supplement attribute the “non-seawater-like REE patterns” to early diagenesis but it is also reasonable to consider that these are non-seawater-like because they are in fact recording a syndepositional signal of poorly mixed fluids. Attributing these patterns to early diagenesis is also contradictory to the supplemental discussion that these samples do not otherwise correlate with other geochemical tests or markers for diagenesis. Therefore, this alternative explanation for the REE patterns (as well as the fluid influencing these carbonates) is feasible and should be considered. Given this prior work, the authors should consider the possibility that the basal ~32 m of section should not be interpreted in the context of global secular seawater evolution. If this were to be the case, what are the impacts on the overall interpretation of environmental change after the Sturtian snowball?

This is a good point. To our knowledge, the above-mentioned hypothesis of ‘meltwater plume’ is first proposed by Shields (2005) to explain the global occurrence of unique cap dolostone after the Marinoan glaciation, which was deposited during the transgressive system tract (TST) and is characterised by tube-like structures and giant wave ripples (Hoffman et al., 2017). However, the TST is absent or highly condensed in non-glacial Cryogenian carbonates directly overlying the Sturtian diamictites (Hoffman et al., 2017). The sedimentary features similar to Marinoan cap carbonates have also not been observed in the studied

interval of Taishir Formation. From the perspective of geochemical signals, the meltwater-affected Marinoan cap carbonates are characterised by unusually high $^{87}\text{Sr}/^{86}\text{Sr}$ values due to the highly radiogenic feature of meltwater (Liu et al., 2014). Similar phenomenon has been observed from the very basal part (~3m) of the Twitya Formation overlying Rapitan diamictites in northwest Canada (Rooney et al., 2014). This observation indicates the possible existence of locally preserved Sturtian cap carbonates. However, we do not observe such high $^{87}\text{Sr}/^{86}\text{Sr}$ values from the basal part of the studied interval in Mongolia (Fig. 2), which are instead consistent with the values from Twitya carbonates (above the highly radiogenic interval; Rooney et al., 2014) and Rasthof carbonates (Yoshioka et al., 2003). These values are now widely regarded as representing the seawater composition (Bold et al., 2016; Cox et al., 2016; Zhou et al., 2020; Bowyer et al., 2022). Together, we suggest that the studied interval has not been affected by the meltwater during deposition and so our interpretation would not be influenced.

On the other hand, we would like to clarify that not all samples from the lower part are characterised by non-seawater-like REE patterns (Fig. S6), which is contrary to what would be expected if a stable meltwater lid impacted the carbonate deposition. Attributing the non-seawater-like REE patterns to meltwater is also inconsistent with the Sr isotope data, as discussed above. Indeed, we have discussed the origin of these non-seawater like REE patterns in detail in our prior study (Zhang and Shields, 2023), and we reached a conclusion similar to that of Lau and Hardisty (2022): proxies' sensitivities to diagenetic processes are variable for different systems and under different local conditions. We showed that REE may be particularly prone to early diagenetic alteration because of the high partition coefficients of REE in carbonate minerals and contrasting REE concentrations in REE carrier phases. It thus may be not surprising that there are no obvious correlations with diagenetic indicators. This is also why we discussed whether other geochemical proxies are altered in conjunction with the distortion of REE in the supplement. It may be interesting to note here that non-seawater-like REE patterns are indeed not uncommon in Neoproterozoic carbonate rocks preserving $\delta^{13}\text{C}$ excursions (e.g., Shuram anomaly), while these rocks are shown to preserve other geochemical proxies that can be globally comparable (e.g., Dodd et al., 2023).

We thank Kimberly for highlighting the potential influence of meltwater plume and for making us aware that readers may have the same question. We have now mentioned this point in the main text (Line 64-67) and added a new section in the Supplementary Information for further clarification.

2. Carbonate U isotope records. The interpretation of the U isotopes is primarily driven by one quite negative $\delta^{238}\text{U}$ value (of $\sim -1.2\text{‰}$ at ~ 32 m). This value is low for carbonates, for this section, and for this time. In addition to the lower samples not containing a seawater-like REE sample and for the potential for a poorly mixed, stratified ocean to be driving the geochemistry of the basal "Sequence 3", there is also potential for this sample to be an outlier. The supplement discusses that this sample has anomalously high Al and dismisses siliciclastic contamination; I think this reasoning is solid, but it still remains that there is something unusual about this sample. It is possible that this sample is representative of a larger trend, but without further samples from this section it is not possible to interpret this confidently. Moreover, Figure 3 compares the Tsagaan Gorge U isotope record with that from other carbonate sections from this basin in Lau et al. (2017). When combined, the significance of this one sample as well as the proposed negative trend in $\delta^{238}\text{U}$ is less obvious. Why not combine the entire record if the U isotope record in the interpretation if a compilation of global S isotope data are also interpreted in reconstructing redox conditions? How would a broader compilation of data from the Taishir Formation modify the interpretations of anoxia  to a more oxidizing ocean?

Thank you for the comment. As mentioned above, the studied interval was not influenced by the meltwater during deposition, and our independent evaluation of U isotope records indicates that the stratigraphic $\delta^{238}\text{U}$ trends are not governed by diagenetic processes. We have reprocessed and remeasured the sample with the lowest $\delta^{238}\text{U}$ value and obtained the same value within uncertainty. The consistency rules out the possibility of experimental error. As suggested, we then combined the entire carbonate $\delta^{238}\text{U}$ records and used LOESS fitting to interpret the trends (Fig. 3), as with recent published U isotope studies (e.g., Zhang et al., 2022; Remírez et al., 2024). It turns out that despite the apparent gap in the dataset caused

by the lithology change (Fig. 2), the overall decreasing trend in the lower part followed by relatively higher $\delta^{238}\text{U}$ values remains quite robust. Therefore, a broader compilation of data from the Taishir Formation further strengthens our argument for the expansion of seafloor anoxia following Sturtian deglaciation. We have now modified the Section 'Transient ocean deoxygenation' based on the trends derived from integrated U isotope dataset. Detailed changes can be found in Line 148-183. We have also updated the Figure 3, as shown below.

Fig. 3. Conceptual model and biogeochemical indicators following the Sturtian

deglaciation. a, Vigorous weathering induced eutrophication after the Sturtian deglaciation. **b**, Expansion of euxinia on productive margins owing to excess of H₂S over iron. **c**, Continuous atmospheric CO₂ drawdown led to subdued weathering, cooling and shallow seawater oxygenation. In (**a-c**), the thicknesses of arrows indicate the relative magnitude of fluxes, and the sizes of ellipses represent the relative sizes of reservoirs. Abbreviations: OM, organic matter burial; Fe_{py}, pyrite burial; DOC, dissolved organic carbon pool. **d**, Global length of continental arcs (Cao et al., 2017) and mid-ocean ridges (Dutkiewicz et al., 2024; Müller et al., 2022). The blue band indicates the studied interval. **e**, Occurrences of biomarkers (modified after Bowyer et al., 2023). **f**, Possible evolution of atmospheric CO₂ and global mean surface temperature (modified after Bowyer et al., 2023; Le Hir et al., 2009). PAL, present atmospheric level. **g-I**, Compilation of carbon, strontium, sulfur, uranium isotopes and iron speciation records (modified after Bowyer et al., 2023). Solid lines in **g,h,j,k** represent the LOESS (locally estimated scatterplot smoothing) fitting curves, and the shaded area represents 1σ (68%) confidence interval. The dashed lines in **j** indicate the major range of δ³⁴S_{py}. In **I**, each boxplot represents the distribution of global Fe_{py}/Fe_{HR} data in each one-million-year time bin. The light shade of pale red band indicates the second stage. The age model of **e-I** is based on model A of Bowyer et al. (2023) (also see Supplementary Fig. 2 for the alternative age model D).

3. Iodine to calcium+magnesium ratios. Iodine ratios are particularly susceptible to diagenesis and will be readily altered before other carbonate geochemical proxies (see Hardisty et al., 2017, cited in the main text; Lau and Hardisty, 2022 GCA). Importantly, iodine ratios that are below detection are also the result of diagenetic alteration. Therefore, the interpretations of iodine ratios below detection as anoxic (or positioned near the oxycline) are possible but cannot be done with confidence when the alternative of diagenetic alteration is also possible. Using the positive Ce anomaly of the lowermost strata to support the interpretations of anoxia isn't convincing because both myself and the authors state that the REE patterns are not representative of seawater. In addition, low I/(Ca+Mg) may also represent a redox stratified ocean and not just local anoxia. The reason for this is because the kinetics of iodide to iodate oxidation are slow, so an ocean poorly oxygenated bottom waters will not be able to produce

a large iodate reservoir (Hardisty et al., 2021 EPSL).

Thank you for the comment. We agree that the carbonate-associated iodine is especially prone to diagenesis. The interpretation of low $I/(Ca+Mg)$ is thus always plagued by the possibility of iodine loss during diagenetic processes. Our samples are characterised by very high Sr concentrations and so they were likely originally aragonites and could have experienced neomorphism during early diagenesis. In this context, we acknowledge that diagenetic iodine loss cannot be ruled out entirely in our case. Therefore, we have largely re-written the relevant sections in both the main text and the supplement, discussing the possibility of iodine loss during diagenetic processes. Nevertheless, as pointed out by Kimberly, the alteration of $I/(Ca+Mg)$ does not necessarily indicate alteration of other proxies and so this would not affect our conclusions. Again, we would like to clarify that not all samples from the lower part is characterised by non-seawater-like REE patterns (Fig. S6). The potential influence of meltwater has also been excluded as discussed above. Therefore, the interpretation of locally reducing environments is in agreement with positive Ce anomalies, which is also consistent with the previous interpretation of Ce anomalies on the same depositional unit from different sections (Lau et al., 2017). Detailed changes can be found in Line 87-106.

3. Age model and correlation of the Taishir Formation to the global compilation. The overall interpretation of the new geochemical data presented here (Figure 3) relies on comparison to the global S isotope compilation, the Fe speciation compilation, the biomarker compilation, and modeled climate data--none of which are from Mongolia. It should be noted that the age model discussion in the supplement proposes that the time interval covered by the studied section is 2-4 Myr (line 83), which is well shorter than the six million years shown in Figure 3.

Comparing records from different sites (e.g., Mongolia to other sites) should be done cautiously as it relies on a highly speculative correlation and age model. The authors acknowledge the challenge here in the supplement, and for correlating Mongolia specifically (lines 68-85). The timing of deglaciation is potentially diachronous (Rooney et al., 2020

Geology) and the duration of the carbonate units within the interglacial period are very difficult to constrain. The age model proposed by Bowyer et al. and presented in Figure 3 proposes that the duration of the T1 member is ~two million years. Based on C isotope chemostratigraphy of a unit that may not be representative of seawater, the age here is not well constrained given the thickness of the unit relative to T2 and T3 and the potential for the cap carbonate at the base of T1 to represent an unusually high sedimentation rate. The challenge of correlating Cryogenian units is certainly not unique to this study. However, because the interpretations combine S isotopes from elsewhere and the implications of this study are interpreted with other geochemical, biomarker, climate, and tectonic data collected elsewhere, additional nuance should be added to the conceptual model to account for this uncertainty.

Thank you for the comment and suggestion. The age model would only be speculative if deglaciation was considered diachronous, but there is little evidence for this. The above-mentioned reference Rooney et al. (2020) indeed summarised available radiometric age constraints and suggested that '*deglaciation was globally synchronous*'. As discussed above, there is also no evidence to suggest that the studied interval represents cap carbonates. The unit thickness could be helpful on estimating duration, but this should also be built on the consideration of lithology. Specifically, T1 is dominated by fine-grain laminated limestone, whereas T2 & T3 are mainly composed of coarse-grain limestone with the presence of ooids. It is reasonable to suggest that sedimentation rates for T2 & T3 are different from and probably higher than that of T1. Hence, the relative thin thickness of T1 could indicate a relatively condensed deposition rather than a short duration. Nevertheless, our estimation assuming constant sediment accumulation rates could provide constraints on the magnitude of duration, which is our main purpose. The estimation suggests that the deposition of studied interval spanned several million years. This estimation agrees with the age model of Bowyer et al. (2023), and hence, we have invoked their age model A in Figure 3 to facilitate the comparison with global data.

The feasibility of the age model can also be tested independently with Sr isotopes (please

see Zhou et al. (2020) for an example of its application). Basically, one can use the well characterised seawater $^{87}\text{S}/^{86}\text{Sr}$ change rate in the Phanerozoic (McArthur et al., 2012) to estimate the duration. The underlying assumption is that the oceanic Sr residence time has not been radically different through the Neoproterozoic to the Phanerozoic, which is likely valid. The magnitude of $^{87}\text{S}/^{86}\text{Sr}$ change in the T1 is ~ 0.0004 . If we take the maximum change rate of $^{87}\text{S}/^{86}\text{Sr}$ during the Phanerozoic, i.e., 0.000192, then the estimated duration should be ~ 2 Myr. Certainly, this estimation carries some uncertainties, but consistent results across different approaches support the conclusion that the deposition of studied interval spanned several million years. We have now added detailed description in the supplement section 'Geological, stratigraphic context and age model'.

We agree that any age model, particularly in deep time, comes with its own uncertainties. In this study, we used the chronostratigraphic framework of Bowyer et al. (2023), which is the best age model we can currently obtain. The way they constructed the age model can be found in their Method section. We note that the Mongolian data have been central to the construction of the model from the beginning. They have also thoroughly discussed the alternative possibilities and limitations of the age model in their supplement. Importantly, they showed that the overall geochemical trends and the sequence of the trends relative to fossil occurrences and climatic steady state would remain consistent in spite of the uncertainties. To illustrate this, we have now tested the alternative age model in the supplement (Fig. S2), which represents an extreme case of a non-glacial duration of c. 21 Myr. As shown below, the overall trends maintain, and this alternative model does not significantly impact our interpretation.

Fig. S2. Conceptual model and biogeochemical indicators following the Sturtian deglaciation. **a**, Vigorous weathering induced eutrophication after the Sturtian deglaciation. **b**, Expansion of euxinia on productive margins owing to excess of H₂S over iron. **c**, Continuous atmospheric CO₂ drawdown led to subdued weathering, cooling and shallow seawater oxygenation. In (**a-c**), the thicknesses of arrows indicate the relative magnitude of fluxes, and the sizes of ellipses represent the relative sizes of reservoirs. Abbreviations: OM, organic matter burial; Fe_{py}, pyrite burial; DOC, dissolved organic carbon pool. **d**, Global length of continental arcs (Cao et al., 2017) and mid-ocean ridges (Dutkiewicz et al., 2024; Müller et al., 2022). The

blue band indicates the studied interval. **e**, Occurrences of biomarkers (modified after Bowyer et al., 2023). **f**, Possible evolution of relative O₂ saturation of surface seawater and global mean surface temperature (modified after Bowyer et al., 2023; Le Hir et al., 2009). **g-I**, Compilation of carbon, strontium, sulfur, uranium isotopes and iron speciation records (modified after Bowyer et al., 2023). Solid lines in **g,h,j,k** represent the LOESS (locally estimated scatterplot smoothing) fitting curves, and the shaded area represents 1 σ (68%) confidence interval. The dashed line in **j** indicates the major range of $\delta^{34}\text{S}_{\text{py}}$. In **I**, each boxplot represents the distribution of global Fe_{py}/Fe_{HR} data in each one-million-year time bin. The light shade of pale red band indicates the second stage. The age model of **e-I** is based on model D of Bowyer et al. (2023).

Minor comments

Line 61: “high purity samples” – what does this mean?

It refers to samples with high carbonate contents. We have modified the sentence for clarity.

As a note, the authors discuss that previous interpretation of the U isotope record grouped together the ferruginous and euxinic sinks (Lau et al., 2017). Others have proposed that if suboxic or ferruginous sediments have a more muted isotope fractionation compared to sulfidic sediments, the relatively high d238U in the lower part of the Taishir Formation (compared to the T3 member which is not discussed here) could represent not oxic but suboxic or ferruginous conditions, at the expense of sulfidic sediments. It should be noted that evidence for ferruginous and euxinic conditions having distinct U isotope fractionation is equivocal (e.g., Lau et al., 2022 GCA, Rutledge et al., 2024 GCA). Nonetheless, I do want to emphasize that I am excited that, while this study interprets the extent of oxygenation to be less widespread than in the 2017 paper, the overall trend in less reducing conditions in T2 compared to T3 is also shown here, and the data in T2 are generally well aligned. (Note that T1 was interpreted with caution due to the potential for deglacial and postglacial stratification leading to non-steady-state behavior and large scatter).

We are also reassured to see that our data align well with those of Lau et al. (2017) for T2. Indeed, $\delta^{238}\text{U}$ data from T1 are also generally consistent, except for the very positive $\delta^{238}\text{U}$ values (Fig. 3). It is true that the uranium isotope fractionation under ferruginous conditions could have a large range, the magnitude of which appears to relate to primary productivity (Lau et al., 2022; Rutledge et al., 2024; Gilleaudeau et al., 2025). The work of Kimberly and her colleagues has greatly advanced our understanding of this aspect and has actually inspired our interpretation. We have now re-written the Section 'Transient ocean deoxygenation', where we expanded the discussion on this point and added the above mentioned references.

Figure 3

- Panel J: it is hard to distinguish between literature data and data presented in this study. The colors should be changed.
- Yellow bar is also difficult to see (on both printed PDF and on my screen)

We have now modified Figure 3, as shown above. Thank you for the suggestion.

- Kimberly Lau, Penn State

Reviewer #2

In this manuscript, Zhang et al. present a new multi-proxy carbonate dataset from Mongolia that spans the interval immediately following the Sturtian glaciation. They argue for an initial pulse of eutrophication and widespread anoxia, followed by a return to less reducing but not fully oxygenated ocean conditions. Overall, I find the manuscript to be comprehensive and well-written, and the interpretations are broadly supported by the data. I recommend publication with minor to moderate revisions. I have given detailed line-by-line comments in both the main text and the supplement (attached files), so please follow those closely. Overall, I feel that some more explanations are needed about some specific diagenetic issues such as fractionation of Zn isotopes during incorporation into carbonate and the correlations between $\delta^{34}\text{S}_{\text{CAS}}$ and variables such as [Al] and Sr/Ca. Including a plate of photomicrographs in the supplement would be helpful to the reader. I point out numerous issues such as this in my comments, but overall I am positive about this manuscript and don't have any overarching criticisms that should prevent publication.

We thank the reviewer for the positive and thoughtful comments.

Line 45: "of the"

Revised.

Line 82: How was this determined? Please point the reader to the appropriate supplementary figure and text that shows how this was determined. How was detrital influence on REE patterns assessed? Did you screen samples based on Y/Ho?

Thank you for the comment. We have now modified the sentence for clarity and added relevant discussion in the supplement. The influence of detritus was first minimised by conducting sequential leaching with dilute acetic acid (Methods) and further assessed by the Al concentration and its correlation with the REE content (Fig. S3). Y/Ho is certainly an

important parameter, but since nearly all samples show moderate positive La and Y anomalies, the samples were screened here mainly based on REE pattern parameters including $P_{\text{N}}/Y_{\text{bN}}$, $S_{\text{mN}}/Y_{\text{bN}}$, and BSI (bell shaped index; Tostevin et al., 2016).

Line 99: This needs further explanation. Ce anomalies show oxic conditions, but $I/(Ca+Mg)$ does not because the site was too close to the oxycline to have appreciable iodate in the water column? Please explain.

The reviewer is correct. The reason relates to the oxidation-reduction kinetics of iodine: while iodate reduction is considered to be rapid, iodide oxidation is relatively slow even with high $[O_2]$ in ambient waters (Chance et al., 2014). Therefore, it has been proposed that low $I/(Ca+Mg)$ values can indicate proximity to the oxycline (Lu et al., 2016; Lu et al., 2018), a point also raised in Kimberly's comment. An example of the decoupling between Ce/Ce^* and $I/(Ca+Mg)$ is the late Paleozoic, during which consistently low $I/(Ca+Mg)$ similar to Proterozoic values has been observed (Lu et al., 2018), but the Ce anomaly during this period shows significantly negative values (Zhang and Shields, 2022). We omitted these explanations in the original manuscript to keep the word count concise and apologize for any confusion this may have caused. However, as mentioned above, we agree with Kimberly that diagenetic iodine loss cannot be ruled out entirely in our case. Hence, we have revised the relevant discussion about $I/(Ca+Mg)$ in both the main text and supplement, but we would like to emphasize that our interpretation of other proxies and our overall conclusion remain unchanged.

Line 104: Fig.2 I think this figure needs a proper stratigraphic log showing the sedimentology and facies of the succession to compare with the geochemical data. Also, why are there only 4 points for $d_{34}SCAS$? And maybe some more explanation of the dashed lines is needed in the caption. For example, why are you presenting the median Archean value for $P/(Ca+Mg)$ when these rocks are Cryogenian? Lastly, it may be best to plot a dashed line in the $d_{238}U$ panel showing the average $d_{238}U$ value of modern carbonates as opposed to modern seawater because of the potential offset in $d_{238}U$ between carbonates and seawater. best

to compare carbonates to carbonates.

Thank you for the suggestion. We have now added the stratigraphic column, showed the dash line for the mean $\delta^{238}\text{U}$ value of Bahamian carbonates (Chen et al., 2018), and changed the caption. There are only 4 $\delta^{34}\text{S}_{\text{CAS}}$ datapoints because only five samples yield sufficient sulfur for isotopic analyses (Methods), with one of them removed from discussion due to the potential of being influenced by pyrite oxidation (Supplement Information). We presented the median $\text{P}/(\text{Ca}+\text{Mg})$ value of Archean carbonates because it represents the first reported statistical result from Precambrian carbonates (Ingalls et al., 2022) and is comparable to our dataset. Nevertheless, our main purpose is to visually separate different data groups, and we have updated the dashed lines accordingly. The revised Figure 2 is shown below.

Fig. 2. Geochemical data profiles for the studied succession from the lower part of Taishir Formation at Tsagaan Gorge. Sequence stratigraphy is from Lindsay et al. (1996). Lithostratigraphy, $\delta^{13}\text{C}_{\text{carb}}$ and $^{87}\text{Sr}/^{86}\text{Sr}$ data (circles) are from Shields et al. (2002). $^{87}\text{Sr}/^{86}\text{Sr}$ data (squares) of Bold et al. (2016) are correlated based on the age model of Bowyer et al. (2023). The mean $\delta^{66}\text{Zn}$ value of the modern deep ocean is from Lemaitre et al. (2020) and the mean $\delta^{238}\text{U}$ value of Bahamian carbonates is from Chen et al. (2018). The shaded band highlights the lower part of the succession. Note that $\text{I}/(\text{Ca}+\text{Mg})$ values below the detection limit are regarded as zero, and errors for some measurements of $\delta^{238}\text{U}$ are smaller than the data symbols.

Line 147: Important to also note that the measured d238U value of -0.43 per mil is still much lower than modern carbonates, which average -0.15 per mil. The carbonate values of -0.43 are similar to most values seen in the mid-Proterozoic as well (Gilleaudeau et al., 2019, EPSL), which may be worth mentioning because it could be a similar interpretation.

Thank you for the comment. We have now added the reference and relevant discussion to the revised manuscript. Detailed changes can be found in Line 164-183.

Line 150: Why would it be weakly euxinic? You have d238U values less than -1 per mil and you have very heavy d34SCAS values, so euxinia must have been expansive. Why are you calling it weakly euxinic?

The inference of a weakly euxinic condition is primarily based on the study of Clarkson et al. (2023). Here weakly euxinic refers to water column $[H_2S_{total}] < 100 \mu\text{mol/mol}$, rather than the spatial extent of euxinia. They found that U isotope fractionation could be very high ($\Delta^{238}\text{U} > +0.6\text{‰}$) under such a condition and so they suggested that this condition could potentially explain the very low seawater $\delta^{238}\text{U}$ values observed during the early Tonian and the late Ediacaran. The weakly euxinic condition also appears to be consistent with the background of low-sulfate seawater in our case. However, we find the term somewhat misleading and have therefore removed it.

Line 159-160: What about carbonates? What is the offset between seawater and carbonate d66Zn? This is critical to discuss as you are presenting carbonate d66Zn data.

Thank you for the comment. Carbonate rocks are estimated to represent a minor sink for Zn (Little et al., 2014; Dickson, 2022). Biogenic carbonates such as corals and foraminifera appear to be able to record the seawater $\delta^{66}\text{Zn}$ values (Little et al., 2021; Druce et al., 2022), inorganic carbonates generally prefer the heavy Zn isotope with an offset of around 0.4‰ potentially due to Zn^{2+} incorporation through tetrahedral coordination (Mavromatis et al., 2019; Müsing et al., 2022; Dickson, 2022). We have now added the relevant content in Line 209-

212.

Line 180-181: But looking at Fig. 2, Zn concentrations are lower in the lower part of the succession compared to the upper part, so couldn't this be because of widespread euxinic Zn drawdown?

This is a good point. It is true that [Zn] is relatively lower in the lower part of the succession, but it does not decrease upsection as would be expected with expanding anoxia/euxinia. Instead, [Zn] shows a consistent increasing trend in the lower part (Fig. 2). Additionally, $\delta^{66}\text{Zn}$ remains relatively stable and low after the initial decrease, despite the expansion of anoxia/euxinia (Fig. 2). These observations lead us to consider it less likely that the expanded euxinic sink is responsible for the $\delta^{66}\text{Zn}$ variation in the lower part. We have expanded the discussion on this point (Line 241-246).

Line 187: But why would more organic-rich sediments be deposited during the less reducing upper interval compared to the strongly anoxic lower interval? For the upper part, you are arguing for less reducing global seafloor conditions but expanded deposition of organic-rich sediments. This seems contradictory. Please explain.

Thank you for the comment. We would like to clarify that we were not comparing the burial flux of organic-rich sediments between the lower and upper parts here. Instead, we focused on the relative importance of the processes that could have caused the $\delta^{66}\text{Zn}$ variation. It is possible that, as the light Zn input from continents gradually decreased as indicated by the flattening of seawater $^{87}\text{Sr}/^{86}\text{Sr}$ curve, the organic matter burial required to counteract the negative $\delta^{66}\text{Zn}$ shift also decreased accordingly. Additionally, the expansion of less reducing conditions could have led to the remobilization of light Zn from continental margin sediments due to the increased dissolved oxygen on the shelf (Sweere et al., 2020). The process, if unaccompanied by increased organic matter burial, would likely lead to the decrease of seawater $\delta^{66}\text{Zn}$ values; but this conflicts with what we observed (Fig. 2). Hence, we think that the inference of relatively increased burial of organic-rich sediments is reasonable. On the

other hand, the change in oceanic redox conditions may be largely related to the shallow ocean (or margin seafloor) as there is no evidence for full ocean oxygenation. Meanwhile, organic matter burial may persist in deeper oceans or upwelling zones, as supported by the consistently high TOC in the Xiangmeng Formation deposited in the basinal environment (Bowyer et al., 2023). Therefore, it appears that relatively increased organic matter burial does not necessarily conflict with the less reducing seafloor conditions. We have modified the content in Line 246-253.

Line 252: Fig.3 How do you explain the different $\delta^{238}\text{U}$ values at very similar times between this study and Lau et al. (2017)? And how was the age model constructed to put these points here on the same plot?

Thank you for the comment. As mentioned in our response to Kimberly, we adopted the age model of Bowyer et al. (2023), which is the best age model we can currently obtain. We have also tested the alternative age model and updated figures (Figs. 3 and S2) as shown above. This information is now clearly mentioned in the updated caption of Figure 3 (Line 339) and in the Supplement Information. Our $\delta^{238}\text{U}$ data are generally consistent with those in Lau et al. (2017) for the upper part (Fig. 3). When combined together, their data also largely follow the decreasing trend in the lower part, except for the very positive $\delta^{238}\text{U}$ values (Fig. 3). It is possible that the uncertainties related to the age model may have caused the different $\delta^{238}\text{U}$ values at very similar times. Alternatively, some noise in the carbonate $\delta^{238}\text{U}$ dataset is quite common in published literature, which has been associated with analytical uncertainties and diagenesis (Kipp and Tissot, 2022), and so the trend is the most important for the interpretation. Particularly, it has been recently suggested that carbonate $\delta^{238}\text{U}$ values $\gg -0.3\text{‰}$ have a strong likelihood of being altered during diagenesis (Kipp and Tissot, 2022; Wang et al., 2022). In this regard, and considering Kimberly's comment about our very negative value being an outlier, one may argue that the two contrasting values at very similar times should both be filtered out. However, this would still not significantly change the trend. Overall, we think that despite some variability in the integrated $\delta^{238}\text{U}$ records, the general trend and, therefore, our interpretation will likely remain valid.

Line 378: Could this be further evidence for sulfate limitation in the oceans at this time?

Yes. We have now mentioned this in Line 179. Thank you for pointing this out.

Supplement Comments

An R-squared of 0.92 between AI and d34SCAS is a strong correlation that is hard to explain away as spurious due to insufficient data. I think you need to discuss what other factors could cause this correlation? I would expect d34SCAS to be most prone to diagenesis of all the carbonate-based proxies you've employed, so I think you can entertain the idea of alteration of d34SCAS signals while still interpreting the other signals are well-preserved.

Thank you for the comment. We also found the positive correlation between $\delta^{34}\text{S}_{\text{CAS}}$ and AI to be perplexing. This is because $\delta^{34}\text{S}$ values of terrestrial reservoirs, except for some carbonates and pyrites, typically fall below +20–30‰ (Hammerli et al., 2021). As such, any increase in clastic-derived sulfur as implied by increasing AI contents would likely decrease rather than increase $\delta^{34}\text{S}$ values in our case. The initial AI data were from leachates prepared for Zn isotopes, which were obtained by dissolving samples in buffered ammonium acetate (pH 5.0). Considering that AI concentrations are highly sensitive to the acid strength and that $\delta^{34}\text{S}_{\text{CAS}}$ data were obtained using 6M HCl leaching, we suspect that the spurious correlation may have been influenced by the AI data. To address this, we redissolved the samples in 6M HCl, measured the major elements and re-plotted them again (Fig. S3). The updated analysis reveals that [AI] no longer shows a significant covariation with $\delta^{34}\text{S}_{\text{CAS}}$ ($R^2=0.35$), as shown below. We have now modified the Supplement Information, updated the Supplementary Figs. 3 and 5, and included the new data in the Supplementary Data.

Fig. S3. Cross plots of geochemical proxies against Al, Fe, Mn and TOC contents. TOC data are from Shields et al. (2002).

Fig. S5. Cross plots of geochemical proxies against commonly used diagenetic indicators.

$\delta^{18}\text{O}$ data are from Shields et al. (2002).

It would be best to present a plate of photomicrographs as a supplementary figure so the reader can see the petrographic textures for himself/herself.

Thank you for the suggestion. The petrographic observations were conducted by the author Graham Shields over 20 years ago, with the findings published in Shields et al. (2002). We utilized the same sample set in this study and have now added the reference to the supplement. Although we made every effort to locate the thin sections, it appears that they have been lost over time. We also checked our samples and unfortunately found no remaining rock pieces suitable for preparing new thin sections, as all of them had been crushed into powder for sulfur isotope analysis.

Repeated word.

Modified.

Again, it is $\delta^{34}\text{S}_{\text{CAS}}$ that is showing potential problems, so I think you need to be open to potential alteration of this signal, which could have occurred while many of the other signals remained well-preserved. $\delta^{34}\text{S}_{\text{CAS}}$ has been shown to be the most easily altered of all the carbonate-based proxies you've analyzed.

We agree that the possibility of a genuine covariation should be considered here, in addition to the potential for a spurious covariation as discussed in the original supplement. Sr/Ca is generally lower in altered carbonates due to the loss of Sr during diagenesis. Therefore, the positive covariation between Sr/Ca and $\delta^{34}\text{S}_{\text{CAS}}$ implies that the relatively lower $\delta^{34}\text{S}_{\text{CAS}}$ values may be the result of diagenetic alteration. It should be noted that we do not observe significant covariations between Mn/Sr, $\delta^{18}\text{O}$ and Sr/Ca, which are expected to covary during diagenesis (Brand and Veizer, 1980). In addition to diagenesis, the variation of carbonate Sr/Ca can also be affected by other factors such as seawater Sr/Ca. Hence, the observed

covariation may also represent closely linked environmental factors that caused the simultaneous variation of $\delta^{34}\text{S}_{\text{CAS}}$ and Sr/Ca. Regardless of the reason behind this covariation, it appears that the extremely high $\delta^{34}\text{S}_{\text{CAS}}$ are unlikely to be significantly altered. Consequently, this would not influence our interpretation of euxinia in the lower part. We have now added the above discussion to the Supplement Information.

I would like to see a stratigraphic plot similar to main text Fig. 2 that shows d18O, Mg/Ca, and Mn/Sr, as well as any other detrital/diagenetic indicators that may be useful. Also, looking at the cross-plots, you have some samples that have very low d18O values less than -10 per mil. These are typically taken to indicate diagenesis. How do you explain those when arguing that the other signals are well-preserved?

Thank you for the suggestion. We have added the figure, as shown below, to the Supplement Information. Carbonate $\delta^{18}\text{O}$ is very sensitive to diagenesis and can be altered at very low fluid/rock ratios. The low $\delta^{18}\text{O}$ values may suggest the alteration of oxygen isotopic system. However, the resistance of different proxies to diagenesis is variable, and the same proxy could exhibit different behaviour under different ambient fluid conditions (Lau and Hardisty, 2022; Zhang and Shields, 2023). Therefore, the alteration of sensitive proxies such as $\delta^{18}\text{O}$ does not necessarily indicate distortion of other signals.

Fig. S4. Chemostratigraphic profiles of carbonate $\delta^{18}\text{O}$, Mg/Ca, Mn/Sr ratios and Sr concentrations for the studied succession from the lower part of Taishir Formation at Tsagaan Gorge. $\delta^{18}\text{O}$ data are from Shields et al. (2002).

More discussion is needed on the criteria you used for identifying seawater-like REE signals.

Thank you for the comment. We have now modified this part accordingly. Detailed changes can be found in the Supplement Information.

OK, thanks for discussing this here. But what about the correlation between d34SCAS and Al and Sr/Ca? I still don't think this have been adequately explained.

Please see our responses above.

Discussion of this is need in the text.

Thank you for the suggestion. We have now incorporated this into the text in the supplement.

References:

- Bold, U., Smith, E.F., Rooney, A.D., Bowring, S.A., Buchwaldt, R., Dudas, F., Ramezani, J., Crowley, J.L., Schrag, D.P., Macdonald, F.A., 2016. Neoproterozoic stratigraphy of the Zavkhan terrane of Mongolia: The backbone for Cryogenian and early Ediacaran chemostratigraphic records. *Am. J. Sci.* 316, 1–63.
- Bowyer, F.T., Krause, A.J., Song, Y., Huang, K.J., Fu, Y., Shen, B., Li, J., Zhu, X.K., Kipp, M.A., van Maldegem, L.M., Brocks, J.J., Shields, G.A., Le Hir, G., Mills, B.J.W., Poulton, S.W., 2023. Biological diversification linked to environmental stabilization following the Sturtian Snowball glaciation. *Sci. Adv.* 9, eadf9999.
- Bowyer, F.T., Zhuravlev, A.Y., Wood, R., Shields, G.A., Zhou, Y., Curtis, A., Poulton, S.W., Condon, D.J., Yang, C., Zhu, M., 2022. Calibrating the temporal and spatial dynamics of the Ediacaran - Cambrian radiation of animals. *Earth-Science Rev.* 225, 103913.
- Brand, U., Veizer, J., 1980. Chemical Diagenesis of a Multicomponent Carbonate System-- 1: Trace Elements. *J. Sediment. Res.* 50, 987–998.
- Cao, W., Lee, C.T.A., Lackey, J.S., 2017. Episodic nature of continental arc activity since 750 Ma: A global compilation. *Earth Planet. Sci. Lett.* 461, 85–95.
- Chance, R., Baker, A.R., Carpenter, L., Jickells, T.D., 2014. The distribution of iodide at the sea surface. *Environ. Sci. Process. Impacts* 16, 1841–1859.
- Chen, X., Romaniello, S.J., Herrmann, A.D., Hardisty, D., Gill, B.C., Anbar, A.D., 2018. Diagenetic effects on uranium isotope fractionation in carbonate sediments from the Bahamas. *Geochim. Cosmochim. Acta* 237, 294–311.
- Clarkson, M.O., Sweere, T.C., Chiu, C.F., Hennekam, R., Bowyer, F., Wood, R.A., 2023. Environmental controls on very high $\delta^{238}\text{U}$ values in reducing sediments: Implications for Neoproterozoic seawater records. *Earth-Science Rev.* 237, 104306.
- Cox, G.M., Halverson, G.P., Stevenson, R.K., Vokaty, M., Poirier, A., Kunzmann, M., Li, Z.X., Denyszyn, S.W., Strauss, J. V., Macdonald, F.A., 2016. Continental flood basalt weathering as a trigger for Neoproterozoic Snowball Earth. *Earth Planet. Sci. Lett.* 446, 89–99.
- Dickson, A.J., 2022. The zinc isotope composition of late Holocene open-ocean marine sediments. *Chem. Geol.* 605, 120971.

- Dodd, M.S., Shi, W., Li, C., Zhang, Z., Cheng, M., Gu, H., Hardisty, D.S., Loyd, S.J., Wallace, M.W., vS. Hood, A., Lamothe, K., Mills, B.J.W., Poulton, S.W., Lyons, T.W., 2023. Uncovering the Ediacaran phosphorus cycle. *Nature* 618, 974–980.
- Druce, M., Stirling, C.H., Bostock, H.C., Rolison, J.M., 2022. Examining the effects of chemical cleaning, leaching, and partial dissolution on zinc and cadmium isotope fractionation in marine carbonates. *Chem. Geol.* 592, 120738.
- Dutkiewicz, A., Merdith, A.S., Collins, A.S., Mather, B., Ilano, L., Zahirovic, S., Müller, R.D., 2024. Duration of Sturtian “Snowball Earth” glaciation linked to exceptionally low mid-ocean ridge outgassing. *Geology* 52, 292–296.
- Gilleaudeau, G.J., Chen, X., Romaniello, S.J., Akam, S.A., Wittkop, C., Katsev, S., Anbar, A.D., Swanner, E.D., 2025. Uranium isotope systematics of a low-productivity ferruginous ocean analog: Implications for the uranium isotope record of early Earth. *Geochim. Cosmochim. Acta*.
- Hammerli, J., Greber, N.D., Martin, L., Bouvier, A.S., Kemp, A.I.S., Fiorentini, M.L., Spangenberg, J.E., Ueno, Y., Schaltegger, U., 2021. Tracing sulfur sources in the crust via SIMS measurements of sulfur isotopes in apatite. *Chem. Geol.* 579, 120242.
- Hoffman, P.F., Abbot, D.S., Ashkenazy, Y., Benn, D.I., Brocks, J.J., Cohen, P.A., Cox, G.M., Creveling, J.R., Donnadieu, Y., Erwin, D.H., Fairchild, I.J., Ferreira, D., Goodman, J.C., Halverson, G.P., Jansen, M.F., Le Hir, G., Love, G.D., Macdonald, F.A., Maloof, A.C., Partin, C.A., Ramstein, G., Rose, B.E.J., Rose, C. V., Sadler, P.M., Tziperman, E., Voigt, A., Warren, S.G., 2017. Snowball Earth climate dynamics and Cryogenian geology-geobiology. *Sci. Adv.* 3, e1600983.
- Ingalls, M., Grotzinger, J.P., Present, T., Rasmussen, B., Fischer, W.W., 2022. Carbonate-Associated Phosphate (CAP) Indicates Elevated Phosphate Availability in Neoproterozoic Shallow Marine Environments. *Geophys. Res. Lett.* 49, e2022GL098100.
- Kipp, M.A., Tissot, F.L.H., 2022. Inverse methods for consistent quantification of seafloor anoxia using uranium isotope data from marine sediments. *Earth Planet. Sci. Lett.* 577, 117240.
- Lau, K. V., Hancock, L.G., Severmann, S., Kuzminov, A., Cole, D.B., Behl, R.J., Planavsky, N.J., Lyons, T.W., 2022. Variable local basin hydrography and productivity control the

uranium isotope paleoredox proxy in anoxic black shales. *Geochim. Cosmochim. Acta* 317, 433–456.

Lau, K. V., Hardisty, D.S., 2022. Modeling the impacts of diagenesis on carbonate paleoredox proxies. *Geochim. Cosmochim. Acta* 337, 123–139.

Lau, K. V., Macdonald, F.A., Maher, K., Payne, J.L., 2017. Uranium isotope evidence for temporary ocean oxygenation in the aftermath of the Sturtian Snowball Earth. *Earth Planet. Sci. Lett.* 458, 282–292.

Le Hir, G., Donnadiou, Y., Godd ris, Y., Pierrehumbert, R.T., Halverson, G.P., Macouin, M., N d lec, A., Ramstein, G., 2009. The snowball Earth aftermath: Exploring the limits of continental weathering processes. *Earth Planet. Sci. Lett.* 277, 453–463.

Lemaitre, N., de Souza, G.F., Archer, C., Wang, R.M., Planquette, H., Sarthou, G., Vance, D., 2020. Pervasive sources of isotopically light zinc in the North Atlantic Ocean. *Earth Planet. Sci. Lett.* 539, 116216.

Lindsay, J.F., Brasier, M.D., Dorjnamjaa, D., Goldring, R., Kruse, P.D., Wood, R.A., 1996. Facies and sequence controls on the appearance of the Cambrian biota in southwestern Mongolia: Implications for the Precambrian-Cambrian boundary. *Geol. Mag.* 133, 417–428.

Little, S.H., Vance, D., Walker-Brown, C., Landing, W.M., 2014. The oceanic mass balance of copper and zinc isotopes, investigated by analysis of their inputs, and outputs to ferromanganese oxide sediments. *Geochim. Cosmochim. Acta* 125, 673–693.

Little, S.H., Wilson, D.J., Rehk mper, M., Adkins, J.F., Robinson, L.F., van de Flierdt, T., 2021. Cold-water corals as archives of seawater Zn and Cu isotopes. *Chem. Geol.* 578.

Liu, C., Wang, Z., Raub, T.D., Macdonald, F.A., Evans, D.A.D., 2014. Neoproterozoic cap-dolostone deposition in stratified glacial meltwater plume. *Earth Planet. Sci. Lett.* 404, 22–32.

Lu, W., Ridgwell, A., Thomas, E., Hardisty, D.S., Luo, G., Algeo, T.J., Saltzman, M.R., Gill, B.C., Shen, Y., Ling, H.-F., Edwards, C.T., Whalen, M.T., Zhou, X., Gutchess, K.M., Jin, L., Rickaby, R.E.M., Jenkyns, H.C., Lyons, T.W., Lenton, T.M., Kump, L.R., Lu, Z., 2018. Late inception of a resiliently oxygenated upper ocean. *Science* 177, 174–177.

Lu, Z., Hoogakker, B.A.A., Hillenbrand, C.-D., Zhou, X., Thomas, E., Gutchess, K.M., Lu, W.,

- Jones, L., Rickaby, R.E.M., 2016. Oxygen depletion recorded in upper waters of the glacial Southern Ocean. *Nat. Commun.* 7, 11146.
- Mavromatis, V., González, A.G., Dietzel, M., Schott, J., 2019. Zinc isotope fractionation during the inorganic precipitation of calcite – Towards a new pH proxy. *Geochim. Cosmochim. Acta* 244, 99–112.
- McArthur, J.M., Howarth, R.J., Shields, G.A., 2012. Strontium isotope stratigraphy. *Geol. Time Scale 2012* 127–144.
- Müller, R.D., Mather, B., Dutkiewicz, A., Keller, T., Merdith, A., Gonzalez, C.M., Gorczyk, W., Zahirovic, S., 2022. Evolution of Earth's tectonic carbon conveyor belt. *Nature* 605, 629–639.
- Müsing, K., Clarkson, M.O., Vance, D., 2022. The meaning of carbonate Zn isotope records: Constraints from a detailed geochemical and isotope study of bulk deep-sea carbonates. *Geochim. Cosmochim. Acta* 324, 26–43.
- Remírez, M.N., Gilleaudeau, G.J., Gan, T., Kipp, M.A., Tissot, F.L.H., Kaufman, A.J., Parente, M., 2024. Carbonate uranium isotopes record global expansion of marine anoxia during the Toarcian Oceanic Anoxic Event. *Proc. Natl. Acad. Sci.* 121, 2017.
- Rooney, A.D., Macdonald, F.A., Strauss, J. V., Dudás, F.Ö., Hallmann, C., Selby, D., 2014. Re-Os geochronology and coupled Os-Sr isotope constraints on the Sturtian snowball Earth. *Proc. Natl. Acad. Sci. U. S. A.* 111, 51–56.
- Rooney, A.D., Yang, C., Condon, D.J., Zhu, M., Macdonald, F.A., 2020. U-Pb and Re-Os geochronology tracks stratigraphic condensation in the Sturtian snowball Earth aftermath. *Geology* 48, 625–629.
- Rutledge, R.L., Gilleaudeau, G.J., Remírez, M.N., Kaufman, A.J., Lyons, T.W., Bates, S., Algeo, T.J., 2024. Productivity and organic carbon loading control uranium isotope behavior in ancient reducing settings: Implications for the paleoredox proxy. *Geochim. Cosmochim. Acta* 368, 197–213.
- Shields, G.A., 2005. Neoproterozoic cap carbonates: A critical appraisal of existing models and the plumeworld hypothesis. *Terra Nov.* 17, 299–310.
- Shields, G.A., Brasier, M.D., Stille, P., Dorjnamjaa, D., 2002. Factors contributing to high $\delta^{13}\text{C}$ values in Cryogenian limestones of western Mongolia. *Earth Planet. Sci. Lett.* 196, 99–

- Sweere, T.C., Dickson, A.J., Jenkyns, H.C., Porcelli, D., Henderson, G.M., 2020. Zinc- and cadmium-isotope evidence for redox-driven perturbations to global micronutrient cycles during Oceanic Anoxic Event 2 (Late Cretaceous). *Earth Planet. Sci. Lett.* 546, 116427.
- Tostevin, R., Shields, G.A., Tarbuck, G.M., He, T., Clarkson, M.O., Wood, R.A., 2016. Effective use of cerium anomalies as a redox proxy in carbonate-dominated marine settings. *Chem. Geol.* 438, 146–162.
- Wang, W. qian, Zhang, F., Shen, S. zhong, Bizzarro, M., Garbelli, C., Zheng, Q. feng, Zhang, Y. chun, Yuan, D. xun, Shi, Y. kun, Cao, M., Dahl, T.W., 2022. Constraining marine anoxia under the extremely oxygenated Permian atmosphere using uranium isotopes in calcitic brachiopods and marine carbonates. *Earth Planet. Sci. Lett.* 594, 117714.
- Yoshioka, H., Asahara, Y., Tojo, B., Kawakami, S. ichi, 2003. Systematic variations in C, O, and Sr isotopes and elemental concentrations in neoproterozoic carbonates in Namibia: Implications for a glacial to interglacial transition. *Precambrian Res.* 124, 69–85.
- Zhang, F., Stockey, R.G., Xiao, S., Shen, S. zhong, Dahl, T.W., Wei, G.Y., Cao, M., Li, Z., Kang, J., Cui, Y., Anbar, A.D., Planavsky, N.J., 2022. Uranium isotope evidence for extensive shallow water anoxia in the early Tonian oceans. *Earth Planet. Sci. Lett.* 583, 117437.
- Zhang, K., Shields, G.A., 2023. Early diagenetic mobilization of rare earth elements and implications for the Ce anomaly as a redox proxy. *Chem. Geol.* 635, 121619.
- Zhang, K., Shields, G.A., 2022. Sedimentary Ce anomalies: Secular change and implications for paleoenvironmental evolution. *Earth-Science Rev.* 229, 104015.
- Zhou, Y., Pogge von Strandmann, P.A.E., Zhu, M., Ling, H., Manning, C., Li, D., He, T., Shields, G.A., 2020. Reconstructing Tonian seawater $87\text{Sr}/86\text{Sr}$ using calcite microspar. *Geology* 48, 462–467.

Response to Reviewer Comments

Reviewer #1

I reviewed the initial submitted manuscript. I appreciate the consideration of my previous comments and concerns, which the authors have done satisfactorily. Specifically, the new discussion in the supplement and in the rebuttal addresses my main comment about the meltwater pulse and I appreciate that the revised manuscript has integrated the data more completely than in the initial version.

I have some minor comments/suggestions prior to acceptance.

We are glad to know that Dr Kimberly Lau is satisfied with our previous revision. We also thank her for taking the time to give constructive comments on our manuscript, which have led to improvement of the work.

Lines 172-175: This is an interesting hypothesis which is made compelling by the tie-in to the productivity trends.

We thank Kimberly for the comment and appreciate the recognition of our interpretation.

Line 179: “in the study samples” or “in these samples”

Thanks, modified.

Line 216: Given that burial of organic matter can result from multiple factors, I suggest connecting organic-rich continental margin sediments to organic carbon flux to productivity more explicitly here.

Thank you for the suggestion. We agree that organic matter burial could relate to other factors in addition to productivity. Nevertheless, our current understanding of Zn isotope systematics

in modern oceans indicates that the direct relationship between $\delta^{66}\text{Zn}$ and productivity remains uncertain (Little et al., 2016; Horner et al., 2021). Therefore, we wrote the sentence this way intentionally to strike a cautious note.

Lines 244-246: can the link between expanded euxinia and $\delta^{66}\text{Zn}$ be made more clear? I didn't quite follow this.

Thank you for the comment. This interpretation or link was first proposed by John et al. (2017). They argued that at steady state, a smaller organic matter burial sink could be compensated for by a greater euxinic sink, which would lead to lower $\delta^{66}\text{Zn}$ values as there would be less isotopically light Zn being removed from the ocean. In our case, however, $\delta^{66}\text{Zn}$ values remain relatively stable despite the further expansion of euxinic conditions, which implies only a limited role for euxinia in controlling $\delta^{66}\text{Zn}$ values through the lower part of the section. We have now rephrased the sentence for clarity (Line 226-228).

Lines 247-249: Perhaps, but what if it's local redox change and not global redox change that is driving the Zn isotope record? It seems like (also?) pulling in the Ce/Ce* would be useful in making this argument.

Thank you for the comment. We have now modified the sentence to incorporate Ce/Ce* evidence (Line 233).

Line 253: This paragraph discusses several mechanisms and trends in Zn isotopes. It is useful but I wasn't clear what the sum of the arguments boiled down to with regard to what the authors are interpreting from the Zn record. It would be useful to add a summary sentence that links the record back to what can be confidently stated about nutrient cycling/availability. i.e. extend the statement about burial flux of isotopically light organic rich sediments outpacing weathering to when and how productivity increased.

This is a good point. We have now added the summary sentence at the end of this paragraph

(Line 238-240). Thank you for the suggestion.

Fig 2. Now some vertical lines are not labeled/explained in caption. I think this was useful in the previous version, even if there was some debate over which threshold was meaningful. That could be explained further in the caption.

Thank you for the suggestion. We have now modified the figure and its caption, as shown below.

Fig. 2. Geochemical data profiles for the studied succession from the lower part of Taishir Formation at Tsagaan Gorge. Sequence stratigraphy is from Lindsay et al. (1996). Lithostratigraphy, $\delta^{13}\text{C}_{\text{carb}}$ and $^{87}\text{Sr}/^{86}\text{Sr}$ data (circles) are from Shields et al. (2002). $^{87}\text{Sr}/^{86}\text{Sr}$ data (squares) of Bold et al. (2016) are correlated based on the age model of Bowyer et al. (2023). The Precambrian $\text{I}/(\text{Ca}+\text{Mg})$ baseline is from Lu et al. (2017). Estimation of seawater sulfate $\delta^{34}\text{S}$ is from Wang et al. (2019). The mean $\delta^{66}\text{Zn}$ value of the modern deep ocean is from Lemaitre et al. (2020) and the mean $\delta^{238}\text{U}$ value of Bahamian carbonates is from Chen et al. (2018). The shaded band highlights the lower part of the succession. Note that $\text{I}/(\text{Ca}+\text{Mg})$ values below the detection limit are regarded as zero, and errors for some measurements of $\delta^{238}\text{U}$ are smaller than the data symbols.

Fig S2 replicates Fig 3?

Thank you for the comment. Fig. S2 represents the output of the alternative age model D. It looks very similar to Fig. 3 because the overall trends remain consistent.

Reviewer #2

The authors have done a nice job revising the manuscript. I do have additional comments that should be addressed in the attached annotated PDF, but they are now relatively minor. Note the comments are in the merged PDF in the portion with the accepted (not tracked) new manuscript. I recommend publication once these few remaining issues are addressed.

We thank the reviewer for the continued positive feedback and constructive comments, which have helped improve the manuscript.

Lines 65-67: This statement lacks context. The reader does not know what you are talking about without looking at the SI. Some more information is needed here to make this stand on its own here in the main text.

Thank you for the suggestion. We have now expanded the relevant content to provide more information (Line 65-74).

Line 146: I think you should add "indicate limited euxinia in an ocean that was still largely anoxic". This will help clarify this alternative interpretation.

Thanks, added.

Line 147: I would change this to "limited".

Thanks, modified.

Line 153: I agree but it is also important to note that this does not require widespread oxygenation. It could have been a reduction in euxinia in a still largely anoxic ocean.

Thank you for the comment. We have now made the point more clearly that the positive $\delta^{238}\text{U}$

shift does not necessarily indicate widespread oxygenation (Line 176).

Line 210: I think you need a few concluding sentences about what your interpretation of the Zn isotope data is. You exclude a lot of potential causes of the Zn isotope variability, but it seems very uncertain and I'm not sure what the final interpretation is. Please summarize in a few sentences.

Thank you for the suggestion. This point was also raised by the other reviewer, and we have now added the summary sentence at the end of the paragraph (Line 238-240).

Line 230: What effect would P availability and primary productivity have on the Zn isotopes?

Thank you for the comment. Increased P availability and primary productivity could enhance organic matter burial, which may in turn drive seawater $\delta^{66}\text{Zn}$ toward higher values if there are no significant changes in other fluxes.

Line: 293-294: This is not clear from the Zn isotope discussion above.

Thank you for the comment. We have now removed this sentence for clarity.

Line 323: I'm still having trouble reconciling this with the Lau et al. (2017) paper that argues for ocean oxygenation in the immediate aftermath of the Sturtian followed by a return to anoxia afterwards. You seem to be arguing the opposite. I still don't think this is adequately addressed in the revised paper. Does Lau's oxygenation interval coincide with your Stage 1? If so, this needs to be explicitly stated to allow readers to make the connection between these 2 papers on the same rocks.

Thank you for the comment. The return to anoxia described in Lau et al. (2017) corresponds to the T3 member of the Taishir Formation, which lies well above the interval investigated in our study. This stratigraphic relationship is illustrated in Fig. S1, which shows the regional

correlation of the Taishir Formation. Conversely, the oxygenation interval proposed by Lau et al. (2017) corresponds to Stage 3 in our study. As detailed in the Section 'Transient ocean deoxygenation', our $\delta^{238}\text{U}$ data align well with those of Lau et al. (2017), but our interpretation differs slightly, reflecting advances in our understanding of U isotope systematics since that publication. Importantly, the episode of deoxygenation identified in the basal part of the Taishir Formation in our study is a novel finding. Professor Lau has confirmed in their review that the $\delta^{238}\text{U}$ records from both studies are broadly consistent and that our new data are not in conflict.